Citation: *Molecular Systems Biology* 9:708
www.molecularsystemsbiology.com

# Metabolic reconstruction identifies strain-specific regulation of virulence in *Toxoplasma gondii*

Carl Song[1,2], Melissa A Chiasson[3], Nirvana Nursimulu[1,4], Stacy S Hung[1,2], James Wasmuth[1,6], Michael E Grigg[3] and John Parkinson[1,2,5,*]

[1] Program in Molecular Structure and Function, The Hospital for Sick Children, Toronto, Ontario, Canada, [2] Department of Molecular Genetics, University of Toronto, Toronto, Ontario, Canada, [3] Molecular Parasitology Section, Laboratory of Parasitic Diseases, NIAID, National Institutes of Health, Bethesda, MD, USA, [4] Department of Computer Science, University of Toronto, Toronto, Ontario, Canada and [5] Department of Biochemistry, University of Toronto, Toronto, Ontario, Canada
[6] Current address: Department of Ecosystem and Public Health, Faculty of Veterinary Medicine, University of Calgary, 3280 Hospital Drive NW, Calgary, Alberta, Canada T2N 4Z6
* Corresponding author. Program in Molecular Structure and Function, The Hospital for Sick Children, 21.9709 Peter Gilgan Center for Research and Learning, 686 Bay Street, Toronto, Ontario, Canada M5G 0A4. Tel.: + 1 416 813 5746; Fax: + 1 416 813 5022; E-mail: john.parkinson@utoronto.ca

Increasingly, metabolic potential is proving to be a critical determinant governing a pathogen's virulence as well as its capacity to expand its host range. To understand the potential contribution of metabolism to strain-specific infectivity differences, we present a constraint-based metabolic model of the opportunistic parasite, *Toxoplasma gondii*. Dominated by three clonal strains (Type I, II, and III demonstrating distinct virulence profiles), *T. gondii* exhibits a remarkably broad host range. Integrating functional genomic data, our model (which we term as *i*CS382) reveals that observed strain-specific differences in growth rates are driven by altered capacities for energy production. We further predict strain-specific differences in drug susceptibilities and validate one of these predictions in a drug-based assay, with a Type I strain demonstrating resistance to inhibitors that are effective against a Type II strain. We propose that these observed differences reflect an evolutionary strategy that allows the parasite to extend its host range, as well as result in a subsequent partitioning into discrete strains that display altered virulence profiles across different hosts, different organs, and even cell types.
*Molecular Systems Biology* **9**: 708; published online 19 November 2013; doi:10.1038/msb.2013.62
*Subject Categories:* metabolic and regulatory networks; microbiology & pathogens
*Keywords:* flux balance analysis; metabolic reconstruction; strain differences; *Toxoplasma gondii*

## Introduction

*Toxoplasma gondii* is an opportunistic single-celled parasite with the capacity to infect any warm blooded animal. Thought to infect one in three people worldwide, infection by *T. gondii* results in toxoplasmosis, typically associated with flu-like symptoms in adults that resolve into a life-long chronic illness. More significantly, *T. gondii* can result in serious ocular disease in healthy adults and may be life threatening for pregnant women, transplant patients, and the immunocompromised (e.g., those living with HIV/AIDS) (Jeannel *et al*, 1988; Luft and Remington, 1992; Wong and Remington, 1994; Belanger *et al*, 1999). Despite its significance, few treatments are available and those that do exist do not promote sterile cure. The situation is further exacerbated with the emergence of new strains of parasites resistant to or tolerant of available prophylactics (Aspinall *et al*, 2002; Djaman *et al*, 2007). Indeed, it is increasingly evident that parasite strain 'Type' is a predictor of virulent disease (Grigg *et al*, 2001b; McLeod *et al*, 2012). Among extant *T. gondii* lines, three strains (referred to as Types I, II, and III) dominate human infections in Europe and North America (Boothroyd and Grigg, 2002). Compared to Type II and Type III, Type I strains display relatively high growth rates and are acutely virulent in mice ($LD_{100} = 1$ parasite) (Howe and Sibley, 1995). Recent work has identified that murine virulence is highly dependent on the expression level of virulence factors, such as ROP18, GRA15, and SRS29C (Melo *et al*, 2011; Wasmuth *et al*, 2012), proteins that target host immune signalling pathways. At the same time, due to its importance in providing energy and the basic building blocks required for growth, metabolic potential is increasingly being viewed as a critical element governing a pathogen's virulence potential, as well as its ability to survive in infected hosts (McKinney *et al*, 2000; Olszewski *et al*, 2009; Willger *et al*, 2009; Ensminger *et al*, 2012). Through modulating metabolic capacity, parasites are able to tune growth in response to changes in host environment, offering a potential route to a broad host range. Genome comparisons reveal identical sets of genes encoding enzymes with the same predicted functional roles across the three strains. However, what is not known is how the differential expression of these genes across different *Toxoplasma* strains may influence their growth potential and hence virulence.

Genome-scale metabolic reconstruction has emerged as an effective strategy for systems-based investigations of an organism's metabolic potential, serving to crystallize current knowledge of an organism's metabolism as well as providing a framework for *in silico* investigation (Becker *et al*, 2007; Oberhardt *et al*, 2009; Thiele and Palsson, 2010). Metabolic reconstruction is an iterative process, beginning with the initial creation of a draft metabolic network based on the available enzyme annotation data. Subsequent rounds of simulation and refinement help resolve errors and fill gaps in otherwise incomplete networks (Green and Karp, 2004). With the increasing availability of high quality metabolic reconstructions, a variety of modeling procedures have been developed to analyse how these reconstructions are organized and operate. Arguably the most established method is flux balance analysis (FBA) (Kauffman *et al*, 2003; Lee *et al*, 2006), which solves for a steady-state distribution of reaction fluxes while satisfying *a priori* constraints (Orth *et al*, 2010). During an FBA simulation, the algorithm identifies sets of metabolic fluxes that optimize a specified function, for example, maximizing growth potential. This is achieved through the derivation of a 'biomass equation', which details the proportions of all metabolites required for growth (e.g., DNA, RNA, protein, lipids, and cofactors). In the absence of constraints within the system, FBA can yield a range of optimal pathways to achieve maximal growth. To reduce the number of pathways, constraints can be placed on individual fluxes. Ideally these are obtained through systematic surveys of enzyme activities. However, in the absence of such data, mRNA expression data have been found to be an effective substitute (Colijn *et al*, 2009; Huthmacher *et al*, 2010). FBA has been successfully applied to a number of pathogens including *Mycobacterium tuberculosis*, *Leishmania major*, and *Plasmodium falciparum* (Raman *et al*, 2005; Chavali *et al*, 2008; Plata *et al*, 2010) to predict enzymes critical for growth and virulence.

Motivated by the need for a clearer understanding of the relationship between strain-specific metabolic capacity and a pathogen's ability to replicate and cause disease across a broad range of intermediate hosts, we present the first high quality metabolic reconstruction and constraint-based model of *T. gondii*. Our reconstruction, termed as *i*CS382, consists of 382 gene annotations, 282 enzymes, 384 unique metabolites, and 571 reactions, and provides a valuable reference resource detailing current state of knowledge of *T. gondii* metabolism. Integrating mRNA expression data, we use this model to identify strain-specific differences in metabolic potential that correlate with their observed growth rates. Drug inhibition assays are then applied to three enzymes to validate findings from our model.

## Results

### A metabolic reconstruction crystallizes our current knowledge of *T. gondii* metabolism

Here, we present a systematic high quality reconstruction of the metabolic capabilities of *T. gondii*. We began by constructing an initial set of 258 enzymatic reactions (unique EC identifiers) that captures: (1) expert curation provided through the established *T. gondii* knowledgebase—ToxoDB (Gajria *et al*, 2008); (2) the curated Braunschweig Enzyme Database (BRENDA) (Barthelmes *et al*, 2007); (3) 43 previously published studies in *T. gondii* metabolism supporting 89 enzymatic reactions; and (4) 132 high confidence predictions of enzymes using the DETECT pipeline (Hung *et al*, 2010) (Figure 1A; Supplementary Table S1). Among these predictions were two genes that appear misannotated in ToxoDB (Supplementary Figure S1). TGME49_088450 is annotated as an aldehyde dehydrogenase (EC 1.2.4.1), but is predicted by DETECT to be 1-pyrroline-5-carboxylate dehydrogenase (EC 1.5.1.12), completing the pathway for autotrophic L-proline biosynthesis from L-glutamate. TGME49_109730 is annotated as a glutathione-disulfide reductase (EC 1.8.1.7), but is predicted by DETECT to be a thioredoxin-disulfide reductase (EC 1.8.1.9), which allows for the regeneration of thioredoxin and consequently the reduction in RNA nucleotides to their DNA counterparts. Note that reactions EC 1.2.4.1 and EC 1.8.1.7 are associated with other genes and are therefore also included in our reconstruction (Supplementary Table S2). During initial FBA simulations (see below), we found it is necessary to include an additional 24 so-called 'gap-filling' enzymes to produce the full complement of biomass components required for parasite growth. Literature surveys were used to assign 282 enzymes (258 curated + 24 gap filling) to five defined subcellular compartments: apicoplast; cytosol; endoplasmic reticulum; mitochondrion and mitochondrial intermembrane space. Some reactions were assigned to more than one compartment; thus 282 unique enzymatic activities (i.e., EC identifiers) were assigned to 400 reactions (Supplementary Table S1). Of these, 352 reactions are predicted through ToxoDB and DETECT to be encoded by 382 genes. Of the remaining 48 reactions, 19 represent 11 unique enzyme activities for which there is biochemical support only (see below), while the remaining 29 reactions are provided by the 24 'gap-filling' enzymes.

In addition to the 400 enzymatic reactions, we included 107 transport reactions comprising 85 organellar transport reactions that shuttle metabolites between different subcellular compartments, as well as 22 extracellular transport reactions that allow metabolites to enter and leave the system. For extracellular transport reactions, 21 represent known transporter proteins or auxotrophies supported by literature evidence, while one (tyrosine transport) represents an auxotrophy predicted by our model. In addition we include seven reactions that allow the diffusion of small molecules previously shown not to require a catalytic transporter (Boyle and Radke, 2009), as well as six sink reactions: metabolite exchanges with no support of prior knowledge but required for the production of defined biomass components. The 85 organellar transport reactions include both those catalyzed by active transporters as well as reactions that permit the passive diffusion of metabolites between different cell locations (Chavali *et al*, 2008; Thiele and Palsson, 2010).

In addition to enzymatic and transport reactions, our reconstruction includes three spontaneous molecular interconversions, three reactions required for recycling of currency metabolites through undefined cofactors, one reaction accounting for oxidative stress and nine reactions representing artificial conversions of biomass components to cell growth

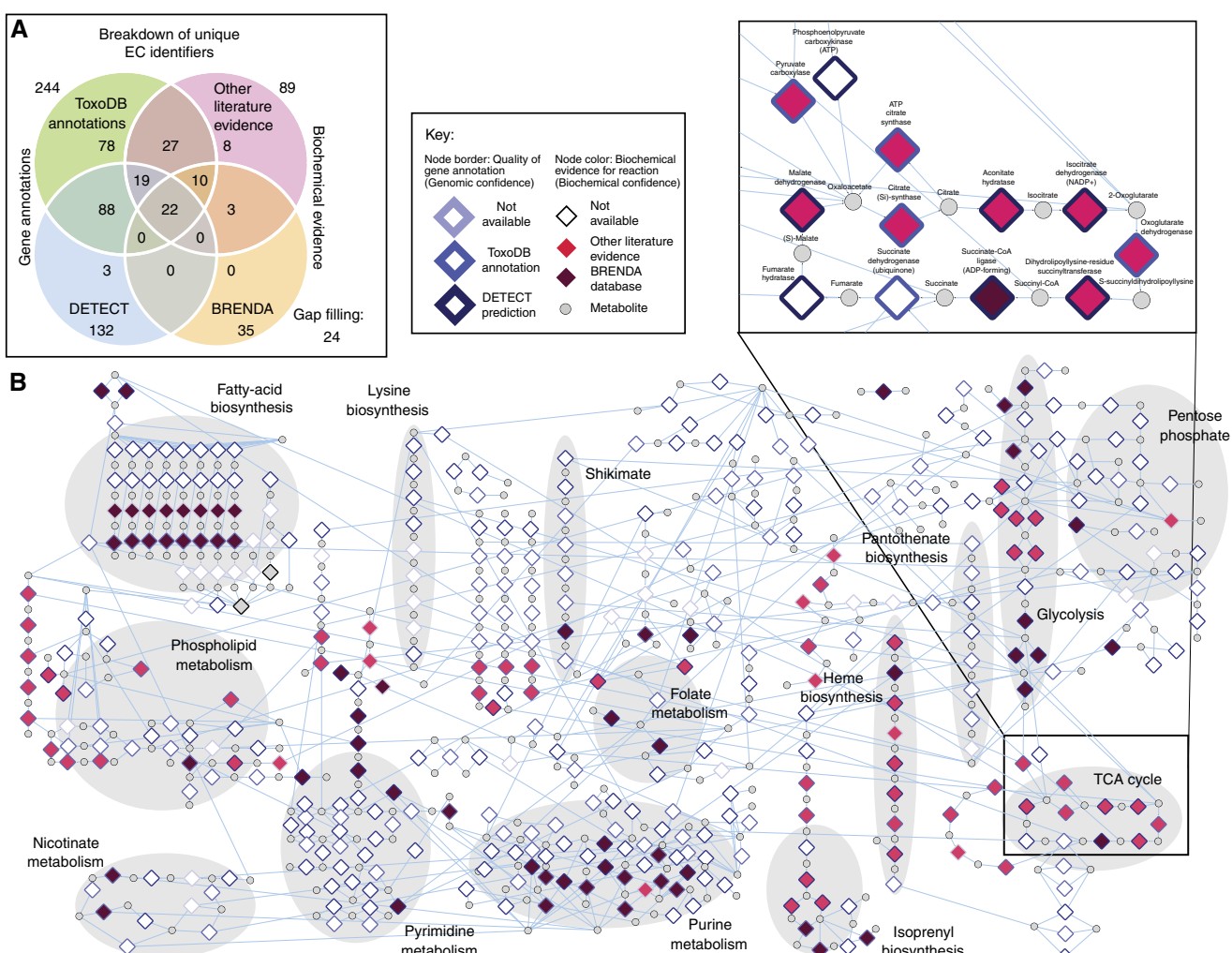

**Figure 1** Overview of *i*CS382. (**A**) Evidence associated with reactions included in the metabolic reconstruction of *T. gondii* (*i*CS382), broken down by source of reaction evidence. Numbers in bold indicate the total number of reactions supported in *i*CS382 by the specific resource. (**B**) Bipartite network representation of *i*CS382 in which nodes represent either reactions (diamonds) or metabolites (circles). Edges between nodes represent enzyme–substrate relationships. Shaded backgrounds indicate groups of enzymes organized into pathways as defined by the Kyoto Encyclopedia of Genes and Genomes (KEGG—Kanehisa *et al*, 2006). A close-up of the TCA cycle showing details of names of enzymes and substrates is also shown as an inset. Reaction nodes are colored according to evidence for gene annotation as well as biochemical data (see inset key). Visualization of the network was performed using Cytoscape (Shannon *et al*, 2003). Edges to common currency metabolites have been removed for clarity.

required by the model. Finally, our model includes an additional 33 so-called 'dead-end reactions', which involve metabolites that are neither produced nor consumed by other metabolic reactions in the network and additionally lack evidence supporting the import or the export of the metabolites implying that the flux for the reaction must be zero. Enzymatic reactions, as defined by EC identifiers, were cross-referenced with the Kyoto Encyclopedia of Genes and Genomes (KEGG) database (Kanehisa *et al*, 2006), to obtain reaction details including substrates and product metabolites, as well as reaction stoichiometry and direction. From an overall set of 571 reactions, the metabolic reconstruction is represented as a network incorporating 492 (384 unique) metabolites (Figure 1B); details of the reconstruction are provided in Table I.

To provide a guide for future hypothesis generation and model refinement, we provide two independent categories of

confidence associated with the annotation of each reaction: *Genomic confidence* indicates that the reaction is supported by a gene association; *Biochemical confidence* indicates that the reaction is supported by experimental evidence (Figure 1B). *Genomic confidence* is assigned on the basis of either DETECT predictions (194 reactions; 132 unique EC IDs) or annotations provided by ToxoDB (348 reactions; 244 unique EC IDs). Similarly, *Biochemical confidence* is assigned on the basis of entries in the BRENDA database (63 reactions; 35 unique EC IDs) or the general literature (126 reactions; 89 unique EC IDs). During the reconstruction, we identified 19 reactions, involving 11 enzymes, for which there is biochemical evidence but for which a gene has yet to be assigned (Table II). These include 2-methylisocitrate dehydratase (EC 4.2.1.99), the only member of the 2-methylcitrate cycle (2-MCC), which generates acetyl CoA from the degradation of branched chain amino acids, for which no gene has currently been assigned

(Seeber *et al*, 2008). Diamine *N*-acetyltransferase (EC 2.3.1.57) is a member of the pathway required for the production of spermidine and putrescine from uptake of host spermine. Intriguingly we note two genes, TGME49_219760 and TGME49_289900, both of which are annotated by ToxoDB as *N*-acetyltransferase family proteins in ToxoDB with the EC identifier corresponding to peptide alpha-*N*-acetyltransferase (2.3.1.88). We propose that since peptide alpha-*N*-acetyltransferase activity is not required by our model, these genes are potential candidates for diamine *N*-acetyltransferase (EC 2.3.1.57) activity. Finally, we note that pyridoxine 5-phosphate synthase (EC 2.6.99.2) consists of a heterodimer composed of two subunits, pdx1 and pdx2, which have been assigned to TGME49_237140 and TGME49_281490, respectively (Muller and Kappes, 2007). Both genes are annotated in ToxoDB as involved in pyridoxal phosphate (Vitamin B6) metabolism but without associated EC identifiers. These examples serve to illustrate the use of metabolic reconstructions to refine future annotation efforts.

Among the six classes of enzymes, transferases represent the most abundant followed by oxidoreductases (Figure 2A). The 492 metabolites involved in these reactions are predicted to be localized in five different subcellular compartments (Figure 2B). The large number of reactions in the cytosol is likely an overestimate as it is the default compartment for reactions without detailed localization data. Figure 2C shows the breakdown of reactions by general metabolic activity. Contrary to a previous metabolic reconstruction of an unrelated unicellular parasite, *L. major*, which highlighted

the importance of lipid and amino-acid metabolism (Chavali *et al*, 2008), *T. gondii*, like *P. falciparum* (Huthmacher *et al*, 2010), possesses a large number of reactions involved in nucleic-acid metabolism. This reflects the impact of purine auxotrophy on apicomplexan parasites, resulting in the retention of multiple purine salvage pathways that can utilize a variety of purine precursors obtained from the host environment (Chaudhary *et al*, 2006). The final model is named '*i*CS382' consistent with metabolic reconstruction naming conventions (Reed *et al*, 2003); further details are provided in Supplementary Table S1. In the next section, we illustrate the use of this model through the application of constraint-based modelling to predict strain-specific metabolic behaviour.

## FBA applied to *i*CS382 reveals strain-specific differences in metabolic potential

Previous studies have shown that *T. gondii* can be classified into three dominant clonal lineages, with each possessing different virulence profiles (Howe and Sibley, 1995). Given the relationship between parasite growth and virulence potential, we examined whether strain-specific differences in metabolic capabilities could account for changes in growth rate and hence virulence. Since genome sequencing reveals identical enzyme complements across strains, any metabolic differences will occur either at the level of gene/protein expression or through sequence variation. Here, we explore the former by integrating gene expression data into a series of FBA simulations to model the metabolic capabilities of the proliferative, tachyzoite form of the parasite across different strains. To our knowledge, this is the first time FBA has been applied to examine the potential impact of enzyme expression across different strains of pathogens.

To determine flux distributions for each reaction, the FBA framework requires a so-called objective function, defined here as maximizing the production of biomass components (i.e., growth; Table III). In the absence of a complete set of biochemical data detailing the kinetics of each *T. gondii* enzyme, flux constraints were assigned on the basis of gene expression profile data (see Materials and methods). These constraints define upper and lower bounds for each reaction. While gene-expression profiles provide only an approximation of enzyme activity, recent studies demonstrate that they significantly enhance model predictions through defining

**Table I** Details of *i*CS382 metabolic reconstruction of *T. gondii* (for further details, see Supplementary Table S1)

| | |
|---|---|
| Genes | 382 |
| Unique EC identifiers | 282 |
| Metabolites | 492 |
| Unique metabolites | 384 |
| Reactions | 571 |
| Enzymatic reactions | 400 |
| Single gene associations | 217 |
| Multiple gene associations | 135 |
| Biochemical evidence only | 19 |
| Gap filling | 29 |
| Transport reactions | 107 |
| Organellar transport | 85 |
| Extracellular transport | 22 |
| Biomass/utility reactions | 31 |
| Dead-ends | 33 |
| Compartments | 5 |

**Table II** List of biochemical reactions in *i*CS382 for which literature evidence exists but for which no gene has yet been assigned

| EC identifier | Enzyme name | Pathway | Reference |
|---|---|---|---|
| 2.1.3.3 | Ornithine carbamoyltransferase | Arginine/Proline | Cook *et al* (2007) |
| 2.3.1.57 | Diamine *N*-acetyltransferase | Polyamines | Cook *et al* (2007) |
| 2.6.99.2 | Pyridoxine 5-phosphate synthase | Misc | Muller and Kappes (2007) |
| 2.7.2.2 | Carbamate kinase | Pyrimidine | Cook *et al* (2007) |
| 3.5.3.6 | Arginine deaminase | Arginine/Proline | Cook *et al* (2007) |
| 3.5.4.2 | Adenine deaminase | Purine | Chaudhary *et al* (2004) |
| 4.2.1.60 | 3-Hydroxydecanoyl-[acp] dehydratase | Fatty-acid synthesis | Dautu *et al* (2008) |
| 4.2.1.61 | 3-Hydroxydecanoyl-[acp] dehydratase | Fatty-acid synthesis | Dautu *et al* (2008) |
| 4.2.1.75 | Uroporphyrinogen-III synthase | Porphyrin | Seeber *et al* (2008) |
| 4.2.1.99 | 2-Methylisocitrate dehydratase | MC cycle | Seeber *et al* (2008) |
| 5.5.1.4 | Inositol-3-phosphate synthase | Phospholipid | Smith *et al* (2007) |

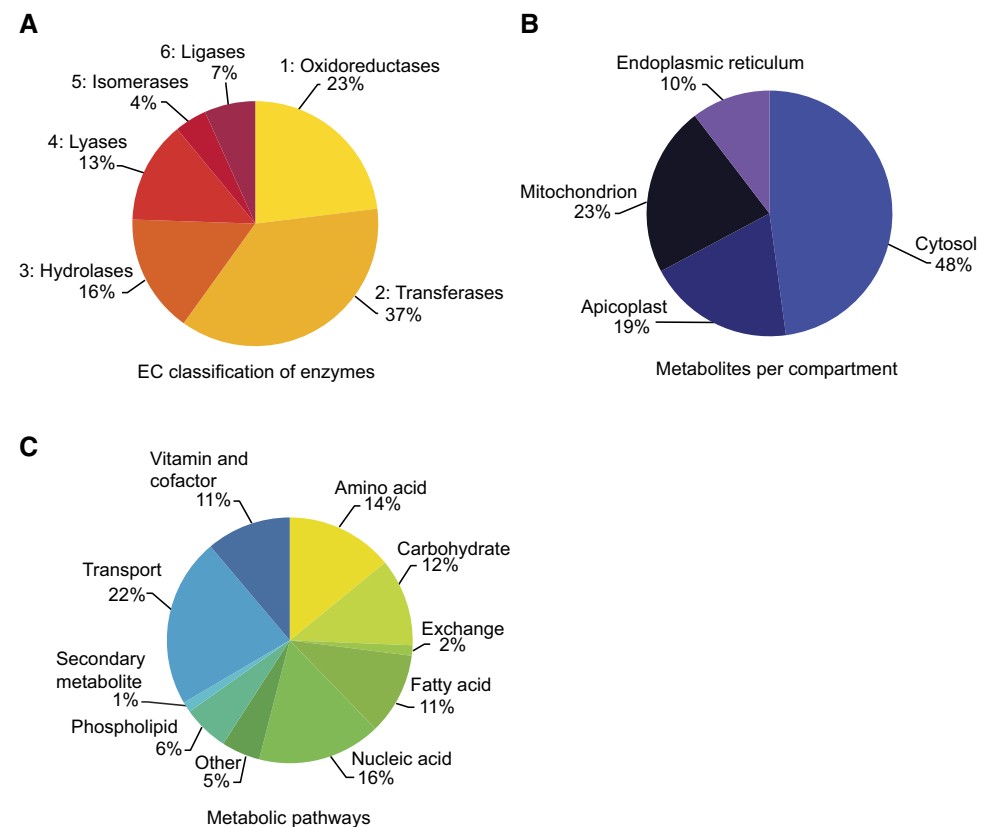

**Figure 2** Statistics associated with metabolic reconstruction of *i*CS382. (**A**) Enzymes in *i*CS382 grouped by enzyme commission (EC) classifications. (**B**) Metabolites in *i*CS382 grouped by organellar compartments. Note a fifth compartment, the mitochondrial inner membrane space, does not contain metabolites and was added to accommodate the H$^+$ gradient required for oxidative phosphorylation. (**C**) Reactions in *i*CS382 grouped by metabolic pathways as defined by KEGG (Kanehisa *et al*, 2006).

maximum flux constraints through reactions (Colijn *et al*, 2009; Plata *et al*, 2010). Our model relies on the assumption that flux associated with an enzyme is dependent on its mRNA expression, for example, strains expressing higher levels of enzyme transcripts will possess a higher flux for that reaction. Consequently, we integrated mRNA expression data previously generated for four strains of *T. gondii* (the Type I strains: RH and GT1, and the two Type II strains: Me49 and Prugniaud) as flux constraints in our FBA model (see Materials and methods). Flux constraints based on the expression data could be added for 209 reactions. These were further scaled to yield a predicted doubling time of 11.8 h for strain Me49 (see Materials and methods). On the basis of the scaling applied to Me49 and consistent with previous experimental studies (Radke *et al*, 2001; Saeij *et al*, 2005), our model correctly predicts that Type I strains of *T. gondii* have much higher growth rates (3.0 and 4.5 h for RH and GT, respectively) than Type II strains (11.8 and 14.0 h for Me49 and Prugniaud, respectively; Figure 3A). On the basis of the expression data, we find that among the pathways upregulated in RH relative to Me49 are pyrimidine biosynthesis, the TCA cycle and the pentose phosphate shunt (Supplementary Figure S2A). These differences suggest that Type I strains produce higher levels of ATP (driving an increased production of biomass) relative to Type II strains. In subsequent simulations, we explore these links in more detail (see below).

It should be appreciated that FBA rarely results in a single solution of optimal reaction fluxes. Nevertheless, by performing flux variability analysis (FVA) (Mahadevan and Schilling, 2003), it is possible to identify reactions that operate only at their maximum constraint for all possible solutions. Such reactions may be deemed 'bottleneck' reactions, since the expectation is that by relaxing the constraint on these reactions, biomass production will similarly increase. We therefore applied FVA to identify potential bottleneck reactions in our four strains that appear to be constrained in different parts of the network (Supplementary Figure S2B). Consistent with our observations on growth rate, four of the eight bottlenecks for the production of biomass for all four strains involve the production of energy. Both ubiquinol-cytochrome-c reductase (EC 1.10.2.2), an essential enzyme in the production of ATP *via* oxidative phosphorylation, and phosphoglycerate kinase (EC 2.7.2.3a and b) are found to be limiting across three strains. Strain RH is predicted to be only constrained by acetyl-CoA carboxylase (EC: 6.4.1.2b), involved in one of the first steps of fatty-acid biosynthesis in the apicoplast. While bottleneck analysis identified the various chokepoints in the model that give rise to the optimal growth rates specific to each strain, it should be appreciated that these findings are reliant on the steady-state assumptions implied in the model, and may therefore not represent the true bottlenecks in a dynamic biological environment. Furthermore,

**Table III** List of biomass constituents with stoichiometric coefficients

| Biomass component | KEGG ID | Coefficient (mmol/g$_{DW}$) | Biomass component | KEGG ID | Coefficient (mmol/g$_{DW}$) |
|---|---|---|---|---|---|
| *DNA* | | | *Lipids* | | |
| dATP | C00131 | 0.002855 | 1-phosphatidyl-1D-myo-inositol | C01194 | 0.007777 |
| dCTP | C00458 | 0.003128 | Phosphatidylcholine | C00157 | 0.08017 |
| dGTP | C00286 | 0.003128 | Phosphatidylserine | C02737 | 0.01084 |
| dTTP | C00459 | 0.002855 | Sphingomyelin | C00550 | 0.01141 |
| | | | Phosphatidylethanolamine | C00350 | 0.0153 |
| *RNA* | | | Cholesterol | C00187 | 0.03525 |
| ATP | C00002 | 0.01678 | Free fatty acids | | 0.252268 |
| CTP | C00063 | 0.02167 | Dodecanoic acid | C02679 | 0.073 |
| GTP | C00044 | 0.02263 | Tetradecanoic acid | C06424 | 0.09 |
| UTP | C00075 | 0.01523 | Palmitic acid | C00249 | 0.166 |
| | | | Stearic acid | C01530 | 0.108 |
| *Proteins* | | | Oleic acid | C00712 | 0.306 |
| L-alanine | C00041 | 0.684 | Linoleic acid | C01595 | 0.198 |
| L-cysteine | C00097 | 0.1297 | Arachidonic acid | C00219 | 0.058 |
| L-aspartate | C00049 | 0.3194 | | | |
| L-glutamate | C00025 | 0.5263 | *Essential molecules* | | |
| L-phenylalanine | C00079 | 0.2276 | Coenzyme A | C00010 | 0.00563 |
| Glycine | C00037 | 0.5002 | Tetrahydrofolate | C00101 | 0.00563 |
| L-histidine | C00135 | 0.1426 | Heme | C00032 | 0.00563 |
| L-isoleucine | C00407 | 0.1603 | Undecaprenyl diphosphate | C04574 | 0.00563 |
| L-lysine | C00047 | 0.2747 | Pyridoxal phosphate | C00018 | 0.00563 |
| L-leucine | C00123 | 0.5988 | Tetrahydrobiopterin | C00272 | 0.00563 |
| L-methionine | C00073 | 0.095 | ADP glucose | C00498 | 0.00563 |
| L-asparagine | C00152 | 0.155 | UDP-*N*-acetyl-D-glucosamine | C00043 | 0.00563 |
| L-proline | C00148 | 0.4197 | GDP mannose | C00096 | 0.00563 |
| L-glutamine | C00064 | 0.2652 | Oxaloacetate | C00036 | 0.00563 |
| L-arginine | C00062 | 0.5445 | Octanoyl-[acp] | C05752 | 0.00563 |
| L-serine | C00065 | 0.7092 | Geranylgeranyl diphosphate | C00353 | 0.00563 |
| L-threonine | C00188 | 0.3327 | | | |
| L-valine | C00183 | 0.3902 | *Growth assoc. maintenance* | | |
| L-tryptophan | C00078 | 0.06729 | ATP + H$_2$O $\rightarrow$ ADP + phosphate | | 28.76 |
| L-tyrosine | C00082 | 0.09781 | | | |

Note that the composition of free fatty acids is used both as a subcomponent of Lipids and as the acyl portion of other phospholipids.

removal of a bottleneck (e.g., by enzyme overexpression) may demonstrate little impact on organism growth *in vitro/in vivo*, as this may simply result in the identification of a new bottleneck reaction restricting growth to a similar (albeit lesser) degree. On the other hand, since these bottlenecks dictate the growth rate of the model, they allow reactions to be prioritized for further *in silico* and *in vitro* experiments that seek to refine associated flux constraints and consequently increase the accuracy of the model.

## FBA of *i*CS382 identifies 39 new potential drug targets

An advantage of *in silico* modelling is the ability to rapidly investigate the impact of gene knockouts. On the basis of the Me49 reconstruction, just less than half of the enzymatic, extracellular, and organellar transport reactions (242 of 507 total) are predicted to be essential for biomass production and may therefore represent suitable targets for therapeutic intervention (Figure 3B and C; Supplementary Table S3). An additional 27 (5.3%) are predicted to have a 'major' impact on growth rate (<80% optimal growth). Of the essential reactions, 171 are encoded by 129 metabolic enzymes. The model correctly predicts the essentiality of dihydrofolate reductase (EC 1.5.1.3) in the folate biosynthetic pathway (targeted by the current anti-*Toxoplasma* therapeutic, pyrimethamine). However, dihydropteroate synthase (EC 2.5.1.15), the target

of sulfadiazine, was not predicted to be essential, presumably due to the presence of thymidylate synthase activity (EC 2.1.1.45) offering an alternative route for the production of dihydrofolate. Further model predictions of essentiality consistent with previous studies include two enzymes involved in fatty-acid synthesis (acetyl-CoA carboxylase, EC 6.4.1.2 and enoyl reductase, EC 1.3.1.9) (Zuther *et al*, 1999; Tipparaju *et al*, 2010). Due to a reliance on purine salvage, adenosine kinase (EC 2.7.1.20) has been suggested as a putative target due to its high level of expression (Rodriguez and Szajnman, 2012). However, the presence of alternative routes for the production of AMP leads the model to predict a non-essential role for this enzyme, with its knockout predicted to have only a modest impact on growth rate (99.6% optimal growth). Finally, contrary to previous experimental studies (Fox and Bzik, 2003), our model does not predict the knockout of carbamoyl phosphate synthetase II (CPSII, EC 6.3.5.5) to impact *Toxoplasma* growth. CPSII catalyses the production of carbamoyl phosphate, a substrate that is acted on by both ornithine carbamoyltransferase (EC 2.1.3.3) and carbamate kinase (EC 2.7.2.2). However, both reactions were set as reversible in the model, allowing them to provide alternate routes to the production of carbamoyl phosphate. But when both reactions were set as irreversible, our model subsequently predicted CPSII to be essential, confirming the experimental data and suggesting that both ornithine carbamoyltransferase and carbamate kinase reactions are irreversible under physiological conditions. In addition, our simulations also correctly predict

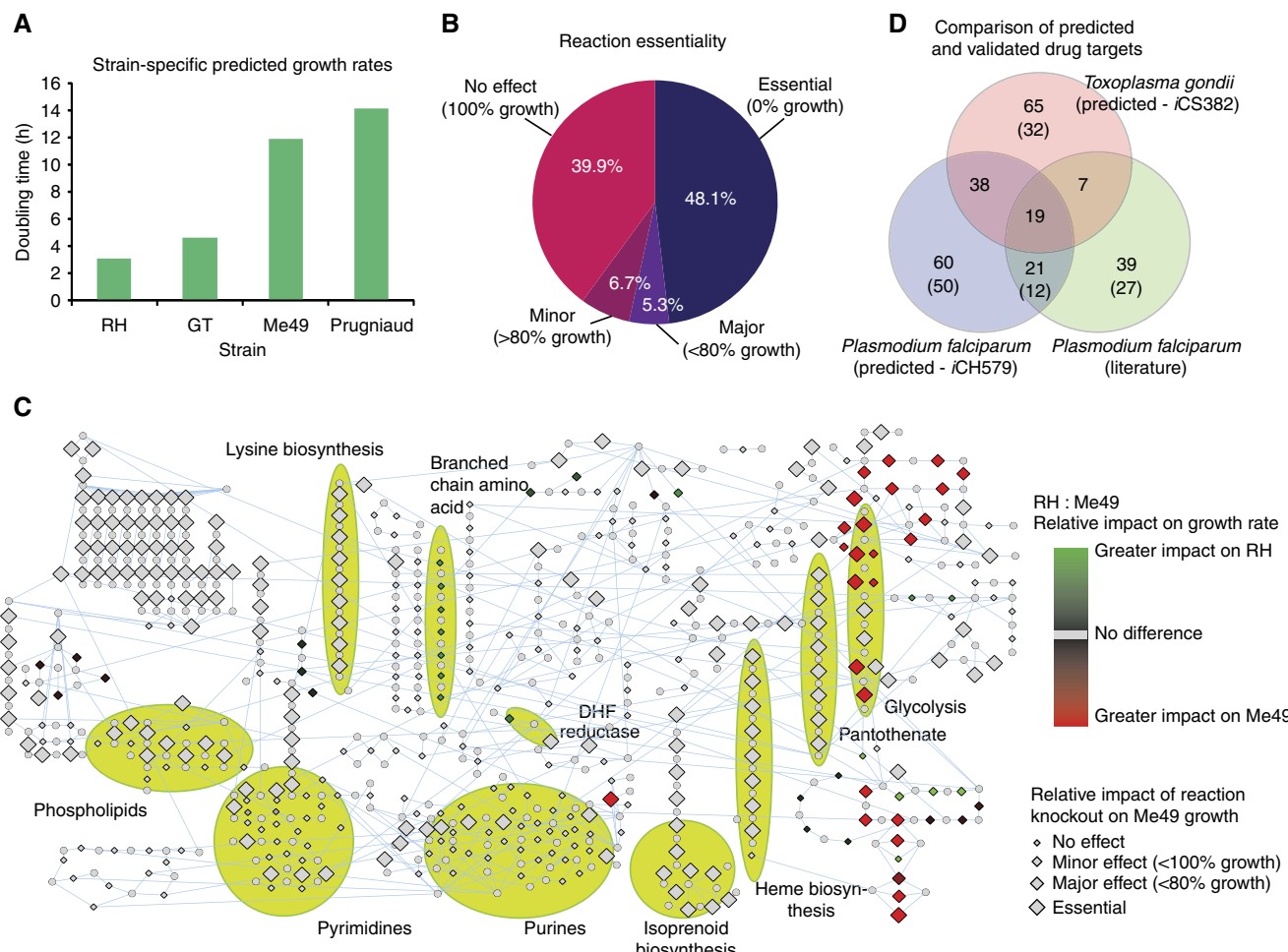

**Figure 3** Strain-specific differences in *T. gondii* metabolism. (**A**) Predicted strain-specific doubling times for four strains of *T. gondii*: Strains RH and GT are Type I parasites, and strains Me49 and Prugniaud are Type II parasites. (**B**) Breakdown of single knockout predictions is classified according to growth ratio with respect to wild type: essential, major effect, minor effect, and no effect. (**C**) Network visualization of predicted impact of reaction knockouts. In addition to identifying reaction knockouts that impact growth of Me49, enzyme reactions (diamond nodes) are colored according to their relative impact between strains Me49 and RH. (**D**) Essential enzymes (EC identifiers) predicted by *i*CS382 compared to *Plasmodium falciparum* model *i*CH579, and a gold standard of biochemically determined essential enzymes. Numbers in brackets show enzymes unique to that species.

tryptophan, cysteine and arginine auxotrophy (Pfefferkorn *et al*, 1986; Fox *et al*, 2004). Contrary to previous findings (Chaudhary and Roos, 2005), our model predicts *Toxoplasma* to be an auxotroph for tyrosine. Recent work has identified a putative phenylalanine hydroxylase (EC 1.14.16.1) (Gaskell *et al*, 2009); inclusion of this reaction results in tyrosine import no longer being essential. These results highlight the iterative process of refining metabolic reconstructions through integration of additional experimental data to increase the accuracy of simulations.

To further prioritize this list, we identified 39 enzymes (unique EC identifiers), catalyzing 67 essential reactions, which lack homologs in humans (Supplementary Table S3). For example, we identified three major biosynthetic pathways essential to the parasite, with enzymes lacking homology in human: isoprenoid biosynthesis, pantothenate biosynthesis, and lysine biosynthesis. Previous studies have already implicated isoprenoid and pantothenate pathways for therapeutic intervention (Muller and Kappes, 2007; Moreno and Li, 2008), however, for lysine biosynthesis, we note mRNA

expression levels of genes encoding five enzymes in the pathway are low, suggesting a reliance on host uptake.

Moving beyond single reaction knockouts, the availability of a robust metabolic model enables the rapid investigation of epistatic interactions within the network (i.e., pairs of reactions with redundant activities). The identification of such reaction pairs offers additional opportunities for therapeutic intervention through targeted drug combination strategies. Ignoring trivial reaction combinations in which one of the reactions is predicted to be essential, we systematically explored the impact of 26 106 pairs of reaction knockouts based on the Me49 reconstruction (Supplementary Table S4). We derived a genetic interaction score, calculated as the ratio of double knockout growth ratio to the product of single knockout growth ratios (see Materials and methods). Of all enzyme pairs simulated for double knockout, 1721 (6.6%) demonstrated any degree of epistasis, of which only 322 (1.2%) were predicted to be lethal (Figure 4). Our results predict that the TCA cycle and the pentose phosphate pathway are functionally redundant for a vital metabolic process,

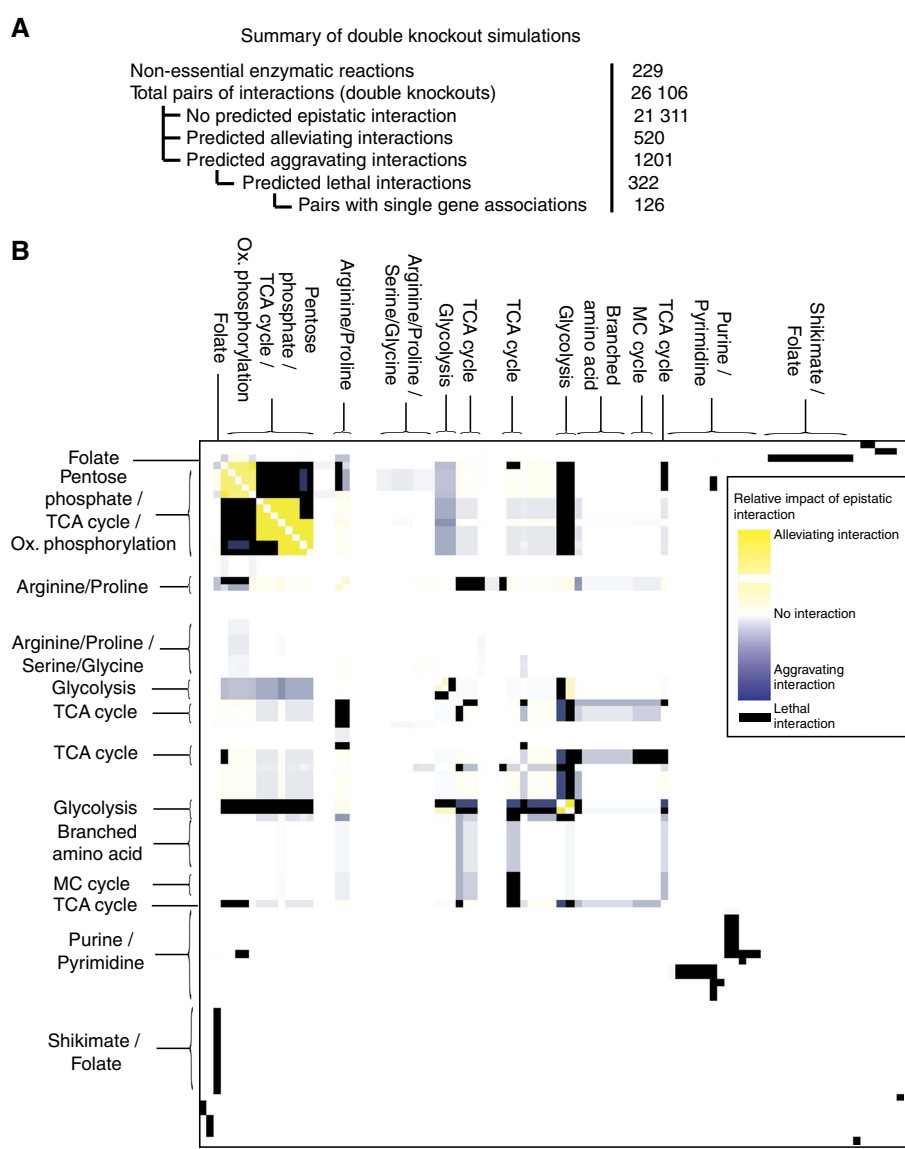

**Figure 4** Epistatic relationships between pairs of enzymes in *i*CS382. (**A**) The total number of reactions paired for double knockout simulations and summary of results. (**B**) Reactions participating in epistatic interactions were clustered on the basis of strength of interaction using cluster 3.0 (average linkage absolute correlation centered) and are visualized as a heatmap.

namely the regeneration of NAD(P)H and the subsequent production of ATP *via* oxidative phosphorylation. Other pathway-centric epistatic interactions of note are the glycolysis pathway in the apicoplast with both the TCA cycle and the pentose phosphate shunt, as well as reactions involved in the inter-conversion of nucleic-acid bases.

## Comparisons of flux balance models identify both species- and strain-specific metabolic dependencies

In previous work, we showed that core metabolic functions encoded by different apicomplexans are provided both by sets of highly conserved enzymes, together with those that are lineage specific (Hung and Parkinson, 2011). The former group of

enzymes is of particular interest as they represent putative pan-apicomplexan therapeutic targets. With the recent availability of a metabolic reconstruction for *Plasmodium falciparum* (Huthmacher *et al*, 2010), as well as a non-exhaustive list of anti-malarial enzyme drug targets determined through previous biochemical studies, we were therefore interested in examining how the different strategies adopted by the parasites to perform similar core metabolic activities might impact reaction essentiality. Figure 3D shows the overlap in predicted essential enzymes (see also Supplementary Figure S2C). Interestingly, while some (64 unique EC identifiers) of the 346 enzymes conserved between *T. gondii* and *P. falciparum* are either predicted (Hung and Parkinson, 2011) or confirmed to be essential in both species, we also note species-specific dependencies. For example, 33 of the conserved enzymes are predicted to be essential only in *T. gondii*, while 31 are either predicted or

**Table IV** Strain-specific differences in response to single reaction knockout

| Reaction | Name | Pathway | Relative growth rate (% of optimal) | | |
|---|---|---|---|---|---|
| | | | RH | Me49 | Difference |
| 5.3.1.1a | Triose-phosphate isomerase | Glycolysis | 75.7 | 7.7 | -68.0 |
| 5.3.1.9 | Glucose-6-phosphate isomerase | Glycolysis | 59.7 | 1.9 | -57.9 |
| 1.2.1.12a | Glyceraldehyde-3-phosphate dehydrogenase | Glycolysis | 54.8 | 0.0 | -54.8 |
| 2.7.2.3a | Phosphoglycerate kinase | Glycolysis | 54.8 | 0.0 | -54.8 |
| 2.7.1.90 | Diphosphate-fructose-6-phosphate 1-phosphotransferase | Glycolysis | 42.5 | 0.0 | -42.5 |
| 4.1.2.13a | Fructose-bisphosphate aldolase | Glycolysis | 42.5 | 0.0 | -42.5 |
| 1.1.1.27 | L-lactate dehydrogenase | Glycolysis | 42.1 | 0.0 | -42.1 |
| T22 | (S)-lactate transport | Transport | 42.1 | 0.0 | -42.1 |
| 2.7.1.40a | Pyruvate kinase | Glycolysis | 41.0 | 0.0 | -41.0 |
| 1.6.2.4 | NADPH-hemoprotein reductase | Ox. Phosphorylation | 73.0 | 40.5 | -32.5 |
| 3.6.3.6 | H+-exporting ATPase | Transport | 31.3 | 0.0 | -31.3 |
| 1.6.5.3 | NADH dehydrogenase (ubiquinone) | Ox. Phosphorylation | 69.5 | 43.7 | -25.8 |
| 1.9.3.1 | Cytochrome-c oxidase | Ox. Phosphorylation | 23.5 | 0.0 | -23.5 |
| 3.6.3.14 | H+-transporting two-sector ATPase | Ox. Phosphorylation | 23.5 | 0.0 | -23.5 |
| 1.1.1.37 | Malate dehydrogenase | TCA Cycle | 86.3 | 63.0 | -23.3 |
| 4.2.1.2 | Fumarate hydratase | TCA Cycle | 86.3 | 63.0 | -23.3 |
| 1.3.5.1 | Succinate dehydrogenase (ubiquinone) | TCA Cycle | 86.3 | 63.0 | -23.3 |
| 5.1.3.1 | Ribulose-phosphate 3-epimerase | Pentose Phosphate | 62.6 | 41.3 | -21.3 |
| 2.2.1.1b | Transketolase | Pentose Phosphate | 62.6 | 41.3 | -21.3 |
| 2.2.1.1a | Transketolase | Pentose Phosphate | 62.6 | 41.3 | -21.3 |
| 2.2.1.2 | Transaldolase | Glycolysis | 62.6 | 41.3 | -21.3 |
| 1.1.1.44 | Phosphogluconate dehydrogenase (decarboxylating) | Pentose Phosphate | 62.3 | 41.1 | -21.2 |
| 1.1.1.49 | Glucose-6-phosphate dehydrogenase | Pentose Phosphate | 62.3 | 41.1 | -21.2 |
| 3.1.1.31 | 6-phosphogluconolactonase | Pentose Phosphate | 62.3 | 41.1 | -21.2 |
| 5.3.1.1b | Triose-phosphate isomerase | Glycolysis | 100.0 | 79.7 | -20.3 |
| 1.2.1.12b | Glyceraldehyde-3-phosphate dehydrogenase | Glycolysis | 100.0 | 79.8 | -20.2 |
| 2.7.2.3b | Phosphoglycerate kinase | Glycolysis | 100.0 | 79.8 | -20.2 |
| 1.10.2.2 | Ubiquinol-cytochrome-c reductase | Ox. Phosphorylation | 55.9 | 43.6 | -12.4 |
| 2.6.1.13 | Ornithine aminotransferase | Arginine/Proline | 72.6 | 82.8 | 10.2 |
| 1.5.1.12b | 1-pyrroline-5-carboxylate dehydrogenase | Arginine/Proline | 72.2 | 82.8 | 10.6 |
| 1.5.1.3b | Dihydrofolate reductase | Folate | 77.1 | 91.9 | 14.8 |
| T5 | Folic acid transport | Transport | 77.1 | 91.9 | 14.8 |
| 1.4.1.4 | Glutamate dehydrogenase (NADP+) | Glutamate/Glutamine | 80.3 | 97.1 | 16.8 |
| 1.2.4.4b | 3-methyl-2-oxobutanoate dehydrogenase | Branched Amino Acid | 82.6 | 100.0 | 17.4 |
| 1.3.99.10 | Isovaleryl-CoA dehydrogenase | Branched Amino Acid | 82.6 | 100.0 | 17.4 |
| 2.3.1.168c | Dihydrolipoyllysine-residue (2-methylpropanoyl)transferase | Branched Amino Acid | 82.6 | 100.0 | 17.4 |
| 2.6.1.42a | Branched-chain-amino-acid transaminase | Branched Amino Acid | 82.6 | 100.0 | 17.4 |
| 4.1.3.4 | Hydroxymethylglutaryl-CoA lyase | Branched Amino Acid | 82.6 | 100.0 | 17.4 |
| 4.2.1.18 | Methylglutaconyl-CoA hydratase | Branched Amino Acid | 82.6 | 100.0 | 17.4 |
| 6.2.1.16 | Acetoacetate-CoA ligase | Misc | 82.6 | 100.0 | 17.4 |
| 6.4.1.4 | Methylcrotonoyl-CoA carboxylase | Branched Amino Acid | 82.6 | 100.0 | 17.4 |
| 2.3.1.9 | Acetyl-CoA C-acetyltransferase | Beta Oxidation | 82.6 | 100.0 | 17.4 |
| T20 | Serine transport | Transport | 73.7 | 97.1 | 23.3 |
| 1.1.1.42 | Isocitrate dehydrogenase (NADP+) | TCA Cycle | 52.9 | 95.8 | 42.9 |
| 4.2.1.3 | Aconitate hydratase | TCA Cycle | 52.9 | 95.8 | 42.9 |
| 1.5.5.1 | Electron-transferring-flavoprotein dehydrogenase | Ox. Phosphorylation | 55.9 | 100.0 | 44.1 |
| 2.3.3.1 | Citrate (Si)-synthase | TCA Cycle | 31.8 | 93.8 | 62.0 |
| 6.4.1.1 | Pyruvate carboxylase | TCA Cycle | 31.1 | 100.0 | 68.9 |

Only enzymatic and extracellular transport reactions with a predicted strain difference of >10% optimal growth are shown.

confirmed to be essential only in *P. falciparum*. Pathways predicted to be essential only in *T. gondii* include lysine biosynthesis and beta-alanine biosynthesis. Of the shared enzymes predicted to be essential in both species are components of glycolysis, pantothenate metabolism, heme biosynthesis, isoprenoid biosynthesis, pyrimidine metabolism, and fatty-acid metabolism—the latter four validated in *P. falciparum*. Due to the conserved nature of predictions, these pathways may be considered as most likely to yield targets for pan-apicomplexan therapeutics.

Moving beyond obvious lifestyle differences at the species level, we were interested in examining how changes in metabolism between different strains of the same species might impact metabolic dependencies. While Type I and Type II strains of *T. gondii* share identical enzyme complements, they possess markedly different growth rates. Here, we hypothesize that differences in enzyme expression result in significant changes in enzyme essentiality. Strains Me49 and RH are predicted to share 228 essential reactions (enzymatic, organellar, extracellular transport and excluding sink reactions), of which 162 are

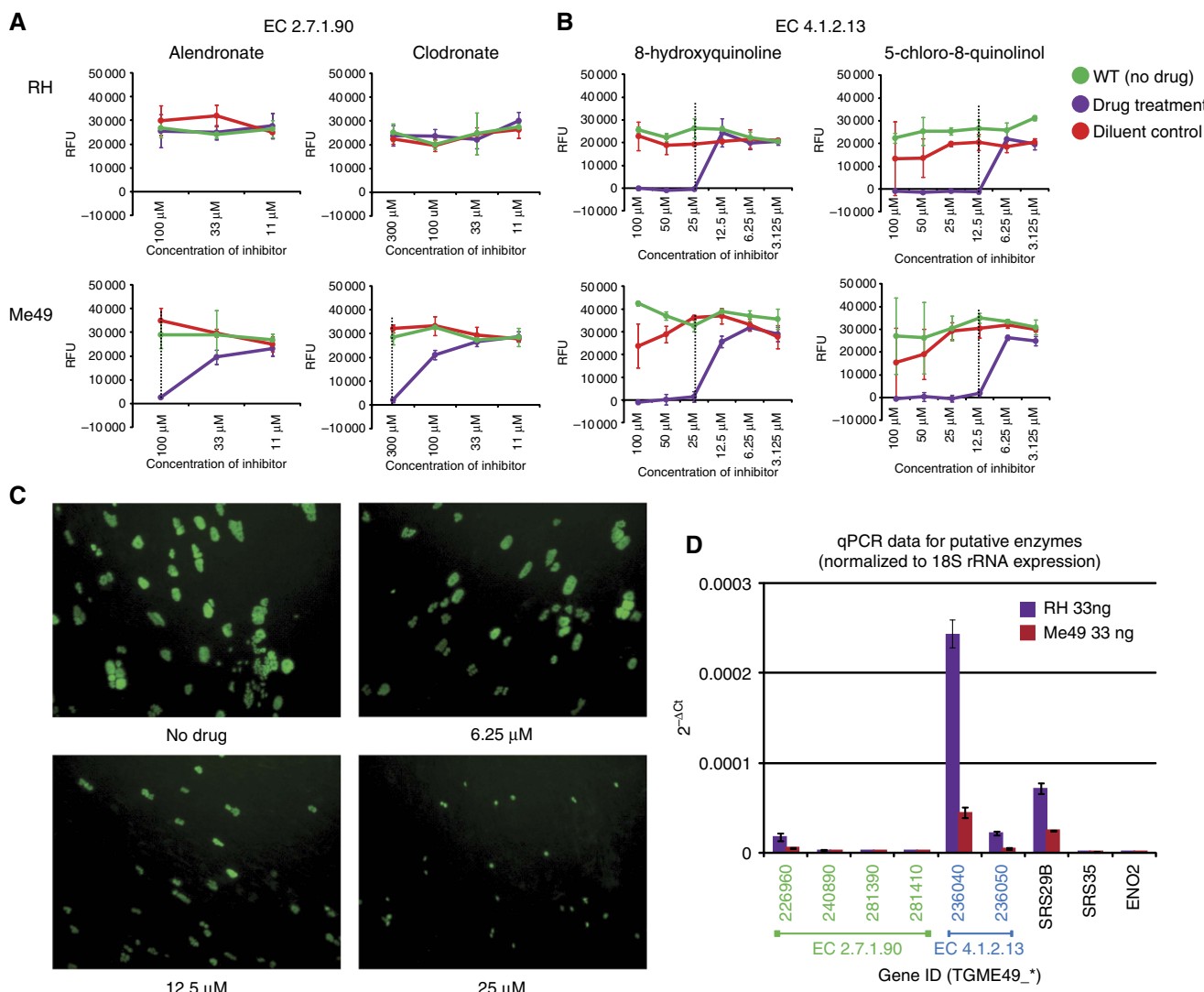

**Figure 5** Impact of four drug candidates on two enzymes involved in glycolysis. (**A**) Seven-day growth assays for Type I (RH) and Type II (Me49) strains under treatment with alendronate and clodronate targeting EC 2.7.1.90. (**B**) Seven-day growth assays for Type I and Type II strains under treatment with 8-hydroxyquinoline and 5-chloro-8-quinolinol targeting EC 4.1.2.13. Red indicates WT (no drug treatment), blue indicates drug treatment, and green indicates diluent control treatment. Dotted lines show lowest molarity at which 100% inhibition of parasite growth occurred. RFUs = relative fluorescent units. (**C**) Micrographs of RH parasites treated for 24 h with 0–25 μM of 8-hydroxyquinoline. (**D**) Expression of target enzymes. To normalize expression we subtracted the 18S Ct value from the Ct value of the gene of interest. For EC 2.7.1.90, only TGME49_226960 is expressed at a significant level. SRS29B serves as a positive control as it is highly expressed in the tachyzoite stage of the parasite. ENO2 serves as a negative control as it is only expressed in the bradyzoite stage.

catalyzed by 121 metabolic enzymes (Supplementary Table S3). A further 14 reactions are predicted to be essential for strain Me49 but not RH. Ten of these involve metabolic enzymes or extracellular transporters, including six encoded by metabolic enzymes associated with glycolysis (Figure 3C; Table IV; Supplementary Table S3). These include lactate dehydrogenase (EC 1.1.1.27) and diphosphate-fructose-6-phosphate 1-phospho-transferase (EC 2.7.1.90), both predicted to reduce growth to 42% of optimal for strain RH. In addition, while not essential to Me49, knockout of glucose-6-phosphate isomerase (EC 5.3.1.9), also involved in glycolysis, was predicted to reduce Me49 growth to 0.2% of optimal compared to 60% of optimal for strain RH. Conversely, our model does not predict any enzymes to be essential for strain RH. Model predictions also identified 18

enzymatic reactions, in which knockout was predicted to result in a significantly greater effect on growth rate (difference of >10% in optimal growth rate between the two strains) for strain Me49 compared to strain RH (Figure 3C; Table IV; Supplementary Table S3). These include enzymes in the pentose phosphate, glycolysis, and TCA cycle pathways. In contrast, 18 enzymatic and 2 extracellular transport reactions were predicted to result in a significantly greater effect on growth rate (difference of >10% in optimal growth rate between the two strains) for strain RH compared to strain Me49 (Figure 3C; Table IV; Supplementary Table S3). These include enzymes in the leucine degradation, oxidative phosphorylation, and TCA cycle pathways. For example, knockout of 3-methyl-2-oxobu-tanoate dehydrogenase (EC 1.2.4.4), hydroxymethylglutaryl-

CoA lyase (EC 4.1.3.4), methylglutaconyl-CoA hydratase (EC 4.2.1.18), and methylcrotonoyl-CoA carboxylase (EC 6.4.1.4) in the leucine degradation pathway were predicted to reduce growth of strain RH to 83% of optimal, with no predicted effect on Me49. Similarly, knockout of citrate synthase, EC 2.3.3.1, was predicted to result in 32% optimal growth for strain RH, compared to 94% optimal growth in strain Me49. Together, these findings indicate how the strains differ in their reliance on energy production pathways, with strain Me49 predicted to be more susceptible to the knockout of enzymes involved in glycolysis compared to RH, while both strains appear to be more reliant on different subsets of enzymes in the pentose phosphate pathway and TCA cycle.

To investigate the robustness of our conclusions and their dependence on the constraints assigned to individual reactions, we performed a systematic set of sensitivity analyses (Supplementary Figure S3). We first investigated the impact of increasing the maximum constraints for each individual metabolic reaction and examined their effect on the single reaction knockout predictions for strain Me49. Of the 400 reactions investigated, changing the constraints of only four reactions had any significant impact on the growth rates predicted for the single reaction knockouts (Supplementary Figure S3A). We also explored the dependence of the observed strain differences in growth rate, on the assigned constraints for eight reactions, mostly involved in energy production pathways (Supplementary Figure S3B). Results from these analyses suggest that our model predictions are robust to changes in the constraints assigned to individual reactions; observed strain differences appear to be driven by relative differences in global enzyme expression across the entire set of energy production pathways.

In the next section, we apply drug intervention screens to validate the observed strain-specific differences for two enzymes involved in glycolysis.

## Drug sensitivity assays validate predicted strain-specific metabolic behavior

The strain-specific differences predicted by the model raise important questions regarding the impact of metabolic regulation on parasite virulence. In an attempt to validate these predictions, we selected three enzyme targets for drug inhibition assays: coproporphyrinogen oxidase (EC 1.3.3.3), fructose-bisphosphate aldolase (EC 4.1.2.13), and diphosphate-fructose-6-phosphate 1-phosphotransferase (EC 2.7.1.90). Coproporphyrinogen oxidase is predicted to be essential for both strains Me49 and RH, while knockout of fructose-bisphosphate aldolase or diphosphate-fructose-6-phosphate 1-phosphotransferase is predicted to be lethal for strain Me49, but to only reduce growth to 42% for strain RH (Table IV). Five drug candidates from a list of 400 possibilities were prioritized based on previous work showing inhibition of growth in other microorganisms (Bai *et al*, 1982; Camadro *et al*, 1986; Bruchhaus *et al*, 1996; Rukseree *et al*, 2008). The five compounds used were 5,5-dithiobis(2-nitrobenzoic acid), alendronate, clodronate, 8-hydroxyquinoline, and 5-chloro-8-quinolinol. Two assays were conducted to measure growth of parasites under drug treatment: a 24-h doubling kinetic assay using microscopy and a 7-day growth assay using fluorescence. Assays at two different time points were necessary; it has been shown that many drugs, including atovaquone and clindamycin, do not act immediately to inhibit growth (Camps *et al*, 2002). Thus, while an inhibitory effect may not be observable at 24 h, at 7 days such an effect should be detectable. To assay for the predicted strain-specific differences in growth, drugs were tested in both RH GFP (Type I) and Me49 RFP (Type II) strains of the parasite.

5,5-dithiobis(2-nitrobenzoic acid) is predicted to inhibit coproporphyrinogen oxidase in the heme pathway (Bogard *et al*, 1989). Under treatment with 5,5-dithiobis(2-nitrobenzoic acid) for 24 h, no inhibitory effect was seen in RH GFP or Me49 RFP. However, after 7 days of treatment, both strains displayed a similar 50% inhibition of growth at a concentration of 900 μM of 5,5-dithiobis(2-nitrobenzoic acid) (Supplementary Figure S4). We next targeted the two enzymes in the glycolytic pathway. First, we applied the bisphosphonate, alendronate, to inhibit the activity of diphosphate-fructose-6-phosphate 1-phosphotransferase (Bruchhaus *et al*, 1996), which our model predicts should have a greater impact on growth of Me49 compared to RH. Consistent with these predictions we found that by day 7, while the growth of RH was not inhibited by alendronate up to a concentration of 100 μM, growth of Me49 was completely inhibited at the same concentration (Figure 5A). The 7-day assay confirmed that alendronate inhibits parasite growth in a strain-specific manner. Previous studies have shown that alendronate can also target the enzyme farnesyl pyrophosphate synthase in the mevalonate pathway (Montalvetti *et al*, 2001). To confirm that the observed strain-specific inhibition was the result of targeting diphosphate-fructose-6-phosphate 1-phosphotransferase in the fructose/mannose pathway, and not a result of off-target effects, we also examined the impact of clodronate (also a known inhibitor of diphosphate-fructose-6-phosphate 1-phosphotransferase; Bruchhaus *et al*, 1996) on parasite growth. Consistent with our findings for alendronate, we observed no effect on RH growth at 7 days up to 300 μM, whereas growth of Me49 was completely inhibited at this concentration (Figure 5A). Finally to determine the importance of the glycolytic pathway on parasite growth, we tested the impact of two additional drugs (8-hydroxyquinoline and 5-chloro-8-quinolinol) that target the enzyme fructose-bisphosphate aldolase immediately downstream of diphosphate-fructose-6-phosphate 1-phosphotransferase (Bai *et al*, 1982; Rukseree *et al*, 2008). After 7 days, treatment with 8-hydroxyquinoline at 25 μM and higher concentrations inhibited growth completely in both RH and Me49 strains, and treatment with 5-chloro-8-quinolinol at 12.5 μM and higher concentrations inhibited growth completely in both RH and Me49 strains (Figure 5B and C). Note parasite growth inhibition is not simply a consequence of the compounds impacting host cell viability (Supplementary Figure S5).

To confirm the relative expression levels of the targeted enzymes, quantitative PCR (qPCR) assays were performed. Four genes (TGME49_226960, 240890, 281390, and 281410) are annotated with diphosphate-fructose-6-phosphate 1-phosphotransferase activity, and are predicted to be targeted by alendronate and clodronate. qPCR revealed that only one of these genes was expressed at an appreciable level, confirming a strain-specific difference in its expression, with RH having significantly higher expression than Me49 (Figure 5D). Two

genes (TGME49_236040 and 236050) are associated with fructose-bisphosphate aldolase activity that 8-hydroxyquinoline and 5-chloro-8-quinolinol are predicted to inhibit; both genes were expressed in a strain-specific manner (although expression of TGME_236040 was greater than TGME_236050) with RH significantly higher than Me49 (Figure 5D). Together, these results demonstrate the usefulness of our metabolic reconstruction, through validating the predicted essentiality of all three enzymes targeted here. In addition, we validated the predicted strain-specific sensitivity of Me49 to inhibition of diphosphate-fructose-6-phosphate 1-phosphotransferase (EC 2.7.1.90) by both alendronate and clondronate. However, we were unable to validate a similar prediction for fructose-bisphosphate aldolase (EC 4.1.2.13). As we note in Discussion, this finding may be related to the translocation of this enzyme during host cell egress (Starnes *et al*, 2006).

## Discussion

There is an increasing recognition that metabolic potential has the capacity to impact pathogen virulence and its ability to expand or restrict its host range (McKinney *et al*, 2000; Olszewski *et al*, 2009; Willger *et al*, 2009). For example, the accumulation of novel mutations in the lysine biosynthetic pathway of the opportunistic pathogen, *Legionella pneumophila*, was recently shown to confer a selective advantage in the mammalian host, at the expense of displayed reduced fitness in its natural host, amoeba (Ensminger *et al*, 2012). In this current study, we have demonstrated that changes in the expression of enzymes in different strains of *T. gondii* can likewise impact parasite virulence, which may impact its ability to expand its host range.

*T. gondii* can be classified into three distinct clonal lineages, designated as Types I, II, and III (McLeod *et al*, 2012). Each type exhibits distinct characteristics in terms of virulence during murine infection. For example, for strain RH; a single parasite is capable of lethal infection in mice, while for strain Me49, the LD50 for mice is 2000 parasites (Howe and Sibley, 1995). Furthermore, different strains exhibit partitioning in terms of their animal host, organs in which they persist and even the host cell type that they target (Wendte *et al*, 2011; Pan *et al*, 2012). Although it is known that different strains differentially target the host immune system (Melo *et al*, 2011), it is still not clear how the biochemical potential of different strains impacts their rate of growth and potential for clonal expansion and success.

To examine variation in metabolic behaviour across the different strains of *T. gondii*, we constructed a manually curated, high quality metabolic reconstruction (*i*CS382) that captures current biochemical knowledge of the parasite's metabolism. Applying a constraint-based framework, we systematically explored strain-specific models based on the differential expression of mRNA data. Although this method does not take into account any post-translational modifications or the specific activity of enzymes, mRNA expression has previously been shown to be useful in determining relative metabolic flux capacity (Colijn *et al*, 2009; Plata *et al*, 2010). Integrating these data we were able to demonstrate a faster growth phenotype for the highly virulent strain RH relative to Me49 consistent with experimental observations (Radke *et al*, 2001; Saeij *et al*, 2005). Our bottleneck and knockout analyses

predicted the increased growth observed for strain RH, which appears due to greater production of ATP through upregulation of enzymes in the glycolytic, pentose phosphate, and TCA cycle pathways. Using the drugs alendronate and clodronate, known inhibitors of pyrophosphate-fructose-6-phosphate 1-phosphotransferase (EC 2.7.1.90), we show a greater inhibition of growth in strain Me49 relative to RH. Interestingly, drug inhibition studies focused on fructose-bisphosphate aldolase (EC 4.1.2.13) did not confirm predictions from our model. This was all the more surprising given that: (1) EC 4.1.2.13 was predicted to be essential for Me49; (2) drug treatment inhibited parasite growth; and (3) qPCR confirmed differential expression of the enzyme between the two strains. To explain this contradiction, we cannot exclude the possibility that *Toxoplasma* possess alternative metabolic routes, not captured in *i*CS382, that render the function of EC 4.1.2.13 redundant. It is also possible that both 8-hydroxyquinoline and 5-chloro-8-quinolinol operate on an alternative target (the two chemical moieties share similar structures). However, a further explanation may concern the translocation of fructose-bisphosphate aldolase (Starnes *et al*, 2006). Specifically, during host cell egress, fructose-bisphosphate aldolase translocates to the parasite's pellicle as a route for optimizing ATP delivery to processes critical for extracellular survival (Pomel *et al*, 2008). Interference with the aldolase enzyme activity has been shown to impact parasite invasion, a necessary stage of the parasite's lifecycle. We therefore speculate that our inability to validate the strain-specific differences predicted by the glycolytic role of the enzyme is masked by the drug's interference post-translocation, impacting the parasite's ability to invade. Nonetheless, the strain-specific behaviour observed in the inhibition of EC 2.7.1.90 demonstrates that while both strains of *T. gondii* possess the same complement of enzymes, changes in their regulation impact not only growth rate, but also the parasite's reliance on critical pathways involved in the production of energy.

These findings may have significant implications for the remarkably broad host range associated with the parasite. *T. gondii* is thought to be capable of infecting any nucleated cell from any warm blooded animal. During this stage of its life cycle, the parasite switches between a proliferative tachyzoite stage and a slow growing tissue cyst bradyzoite stage capable of transmissible infection to other hosts. In order to facilitate its development into the infective bradyzoite form, the parasite must strike a balance between: (a) growing too rapidly such that it kills its host before it has an opportunity to differentiate into the bradyzoite form; and (b) growing too slowly such that the host's immune system sterilizes the infection, again before it has an opportunity to differentiate into the bradyzoite form. Consequently, to be able to successfully colonize a wide range of hosts, the parasite must be able to rapidly modify its growth potential to account for the variety of immune responses and nutrient availability it encounters.

Previous studies suggest that regulatory changes in core components are a major evolutionary force driving the generation of viable selectable phenotypic variation that may operate at a faster rate than the innovation of new genes, which typically occurs through gene duplication and subsequent sub- or neo-functionalization (Carroll, 2000; Wray *et al*, 2003; He and Zhang, 2005; Gerhart and Kirschner, 2007;

Han *et al*, 2009). On the basis of our current findings, we propose that changes in the regulation of enzymes required for the production of energy, as opposed to changes in the enzyme coding sequence itself as observed for *L. pneumophila* (Ensminger *et al*, 2012), offer a rapid evolutionary vehicle that allows *T. gondii* to modify its rate of growth in response to changes in host environment. Hence, observed differences in enzyme expression present the parasite with a fundamental route to adapt to changes in host environment through optimizing exploitation of nutrient availability. Recent studies have shown that sexual crosses of *T. gondii* in the definitive feline host result in the generation of a vast genetic repertoire of progeny (Grigg *et al*, 2001a). As subtle differences in the regulation of metabolic enzyme expression arise, the parasite is able to explore a vast landscape of metabolic potential, each with the possibility of increasing the efficient use of host nutrient availability. At the same time, in areas of restricted host heterogeneity, it might be expected that the parasite's metabolism becomes increasingly optimized for a limited host range, resulting in a subsequent partitioning into discrete strains that display markedly altered virulence profiles across different hosts, different organs, and even cell types (Wendte *et al*, 2011; Pan *et al*, 2012). As additional strains of *T. gondii* are sequenced, we will be able to explore these questions in more detail to examine both the diversity and the extent of metabolic potential that may underlie this drive toward optimizing host exploitation.

Beyond providing insights into the potential for changes in metabolic capacity to drive strain partitioning, *i*CS382 serves as a valuable platform to drive the design of anti-parasitic therapeutics. Studies of *T. gondii* metabolism, capitalizing on genomic data sets, are beginning to identify pathways that may represent additional routes for therapeutic investigations, including the biosynthesis of isoprenoids (Mol and Oudega, 1996), vitamins (Muller and Kappes, 2007), and polyamines (Cook *et al*, 2007). On the other hand, several biosynthetic processes previously thought to be critical for parasite survival have subsequently been found to be complemented by salvage pathways (i.e., scavenged from the host) (Massimine *et al*, 2005; Crawford *et al*, 2006). In a systematic set of single reaction deletion simulations, *i*CS382 predicts 242 of the 507 reactions to be essential for parasite growth including components of the beta-alanine, pantothenate, and glycolytic pathways. Integration of human homology data allows the prioritization of candidates suitable for therapeutic intervention. With predicted differences in essentiality and growth, it is clear that strain-specific differences in chemotherapeutic sensitivity must be considered during the development of any novel therapeutic. This is consistent with a previous study of *Trypanosoma brucei rhodesiense* infections in a murine model, which revealed significant strain-specific responses to drug treatment, with only six of eight clinical isolates susceptible to MDL73811, a potent inhibitor of *S*-adenosyl-methionine decarboxylase (Bacchi *et al*, 1992).

In addition to strain-specific considerations, a valuable goal in these types of investigations is to identify candidates that represent new broad spectrum therapeutic targets. Comparison of our model predictions with those from a previous constraint-based modelling study of *P. falciparum* metabolism (Huthmacher *et al*, 2010) revealed a reasonably high level of overlap in essential reactions (57 of 116 common EC identifiers). Remarkably, despite their relatively close evolutionary relationship, comparisons of enzyme complements between the two reconstructions find only 193 enzymes (of the 609 enzymes included in both reconstructions) in common. Despite this lack of overlap, both models demonstrate the ability to support growth, lending support to the idea that each parasite lineage has evolved distinct metabolic strategies to exploit differences in nutrient availability provided by their unique host profiles (Hung and Parkinson, 2011). For example, while *T. gondii* has the biosynthetic capacity to produce phospholipids as well as several amino acids including serine, alanine, and glycine, *P. falciparum* relies on the breakdown of host heme and scavenging of host lipids. Moving beyond single drug therapies to reduce the rate of development in drug resistance, often a combination of enzyme inhibitors is administered to disrupt pathogen metabolism synergistically. This approach has widely been used for antimalarial and HIV treatments, and has also been shown to overcome drug resistance in *Leishmania* (Perez-Victoria *et al*, 2006). Our model identified 322 combinations that might serve as a useful basis for developing combination therapies in *T. gondii.*

In this study, we present a metabolic reconstruction of *T. gondii* and demonstrate how it might be usefully exploited not only to gain insights into parasite metabolism, but also to inform the choice of new targets for therapeutic intervention. In addition to validating predictions of strain-specific dependencies, this work unveils a host of new testable hypotheses concerning the reliance on specific pathways by different strains as well as a large number of new predicted drug targets. We acknowledge that the more streamlined nature of our *T. gondii* model may reflect the minimalist approach used in the reconstruction process and that there is much potential for future model refinement. These refinements include the incorporation of additional host-scavenging functions, which through the provision of key metabolites has the potential to render otherwise auxotrophic reactions non-essential. In addition, due to the incompleteness of the initial network reconstructed from current knowledge, a number of gap-filling reactions were added to the model to complete essential metabolic pathways. Furthermore, we have identified 33 so-called dead-end reactions that produce metabolites which are neither transported nor used as part of the biomass equation. As additional biochemical investigations are performed, we may expect these reactions to be confirmed and extended or alternative routes to be discovered. Future iterations of the model are also expected to incorporate additional compartments and pathways. For example, we do not currently include either the nucleus or the extracellular milieu as separate compartments. In terms of pathways, while we include a framework for glycan metabolism, the lack of detailed data concerning the role of glycans in the production of biomass for *Toxoplasma* (e.g., abundance and composition of GPI-anchors) has resulted in these pathways being largely incomplete. Similarly, despite evidence of 15-lipoxygenase activity in *Toxoplasma*, lack of additional evidence for supporting reactions precluded the incorporation of a complete arachidonic acid pathway, instead the production of arachidonic acid is captured by a single 'black box' reaction. Finally, it should be appreciated that expression data provide

only a crude approximation for reaction constraints used in the model. Detailed kinetic data, preferably obtained from *in vivo* experiments, would significantly contribute to the accuracy of the model. These caveats aside, by crystallizing current knowledge of *T. gondii* metabolism, we have shown how this resource may be leveraged as a valuable platform for integrating and organizing additional meta-datasets such as mRNA and protein expression data, SNPs and other comparative data sets. To facilitate access to *i*CS382, we make the model reconstruction freely available in the accepted reporting standard, systems biology markup language (SBML) level 2 format, available for download from our project website (http://www.compsysbio.org/projects/iCS382). Our vision is that subsequent investigations will serve to both refine the model itself and shed light on the organization of *T. gondii* metabolism and enzyme dependencies throughout the entire life cycle of the parasite.

# Materials and methods

## Metabolic network reconstruction

An initial list of enzymes was compiled from the following resources: (1) ToxoDB gene annotations (Gajria *et al*, 2008); (2) BRENDA (Barthelmes *et al*, 2007) entries with *T. gondii* annotations; and (3) DETECT predictions (Hung *et al*, 2010) from *T. gondii* gene models (version 2008-07-23). Segregated reactions that do not link to other parts of the network and are responsible for the production of non-essential metabolites or whose substrates are not considered metabolites (e.g., polypeptides or other polymers) were removed. Subsequent manual analysis identified pathways with missing reactions by comparing the network to known pathway schemes in KEGG. A search in the literature was conducted to provide evidence for the presence of the pathway in *T. gondii*. Missing reactions were then added to the pathway if: (1) the pathway was reported or predicted with high confidence in the literature; (2) intermediate metabolites associated with the missing reactions were reported in the literature; or (3) a sizeable fraction of other reactions associated with the pathway were present. Lack of evidence resulted in the pathway being removed. For each enzyme, additional details such as substrate and product metabolites, direction of reaction, subcellular localization, and pathway annotation were retrieved from KEGG (Kanehisa *et al*, 2006), BRENDA (Barthelmes *et al*, 2007), and other literature sources. Intracellular transport reactions were added to move currency metabolites ($H_2O$, ATP, etc.) between compartments. These include both previously known transporter proteins catalyzing intracellular transport reactions for specific metabolites and reactions involving passive diffusion. Negatively charged metabolites translocating to a relatively basic environment (i.e., from cytosol to mitochondrion) were accompanied by hydrolysis of ATP. The reconstruction is maintained as a spreadsheet (Supplementary Table S1) in a standard format (Thiele and Palsson, 2010). Gene to enzyme mappings, metabolite mappings and full citations are listed in Supplementary Tables S2, S5, and S6 respectively. Note here we represent enzymes performing the same reaction in different compartments as different reactions. The metabolic network was visualized as a bipartite graph using Cytoscape (Shannon *et al*, 2003), with nodes representing reactions and metabolites.

## Biomass estimations

The mass of a single *T. gondii* cell together with its protein and lipid composition was obtained from a previous study of membrane fluidity (Gallois *et al*, 1988). The DNA fraction was calculated from the genome size, and the RNA fraction was estimated using a ratio of 1:7 DNA:RNA observed in other organisms (Beste *et al*, 2005). The amino-acid composition of the protein fraction and the NTP composition of RNA were estimated from the codon usage table available on ToxoDB (http://www.toxodb.org), while the dNTP composition of DNA was tabulated from the genomic sequence. For the remaining 3.76% of unaccounted total cell mass was unaccounted for by amino acids, lipids, DNA or RNA and was simply evenly distributed to cofactors and macromolecule subunits essential for parasite growth. Amounts of each biomass component were converted to mmol/$g_{DW}$ (Thiele and Palsson, 2010). We also included two additional biomass terms: growth-associated maintenance (GAM) and non-GAM (NGAM), which define the energy cost in terms of ATP hydrolysis for the organism to grow and sustain, respectively. GAM is estimated from the number of peptide and nucleotide bonds needed to polymerize the DNA, RNA, and protein fractions from their respective subunits. For NGAM, we define an invariable flux of 5 mmol/$g_{DW}$·h (Thiele and Palsson, 2010).

## Flux balance analysis

FBA was performed using the COBRA Toolbox (version 1.3.4) in MATLAB (Becker *et al*, 2007). Each reaction in the model is supplied upper and lower constraints for its flux. By default, irreversible reactions have their fluxes constrained to 0–1000 mmol/$g_{DW}$·h, and reversible reactions −1000 to 1000 mmol/$g_{DW}$·h. For reactions with single-gene associations, mRNA expression values derived from the Gene Expression Omnibus (GEO accession: GSE22315) were used to derive flux constraints. Each reaction receiving a flux constraint based on its associated gene expression relative to the highest gene expression value in the data set (which received an initial constraint of 1000 mmol/$g_{DW}$·h). For irreversible reactions, the lower bound was set to 0 whereas for reversible reactions, the lower bound was set at the negative of the upper bound. For example, a reversible reaction encoded by a gene expressed at 20% of the level of the most highly expressed gene received constraints of −200 to 200 mmol/$g_{DW}$·h. On the other hand, an irreversible reaction encoded by a gene expressed at 20% of the level of the most highly expressed gene received constraints of 0 to 200 mmol/$g_{DW}$·h. The entire set of constraints was then scaled linearly (with the exception of NGAM which was set at a constant flux of 5 mmol/$g_{DW}$·h consistent with previous studies; Thiele and Palsson, 2010) so that the predicted doubling time for Me49 matched the *in vivo* observation of 11.8 h (Blader *et al*, 2001). The calculation of scaling factors was performed only once, for the Me49 expression data. The same scaling of constraints based on the Me49 doubling time was then applied to expression data from the other three strains (RH, GT1, and Prugniaud).

## Bottleneck analysis

For a given FBA solution, the optimization of the objective function is constrained by one or more bottleneck reactions in the network. Such bottlenecks were identified as those reactions with fluxes which equal to their maximum constraint (or minimum, in the case of reversible reactions) for all possible solutions leading to the optimized objective function. The solution space was computed using FVA, part of the COBRA Toolbox package. Reactions in the model were flagged as bottlenecks if the minimum flux of the reaction was equal to its maximum constraint.

## Gene deletion predictions

Single knockouts were simulated for each reaction in the model by setting the constraints of the reaction to 0. Knockout effects were assessed by computing a growth ratio, which is the biomass production rate of the knockout divided by that of the wild type. Non-lethal single knockouts (having non-zero growth ratios) were permutated in pairs for double knockout simulations. Each double knockout simulation is in essence two single knockouts performed in tandem. Double knockouts were assessed by computing a genetic interaction score that is defined as follows:

$$f = gr_{xy}/gr_x gr_y$$

where gr is the growth ratio of the respective knockouts.

## Cell and parasites

Two strain-specific transgenic lines of *Toxoplasma gondii* were engineered. eGFP driven by the GRA1 promoter was stably integrated into the Type I RHΔ*hxgprt* strain whereas DSRed 2.0 (RFP), also driven by the GRA1 promoter, was stably integrated into the Type II Me49 strain by Restriction Enzyme Mediated Insertion (REMI). Parasites with stable integration of the GFP or RFP cassette were selected by fluorescence. The point of insertion is unknown, but the parasites are identical in phenotype to wild type in terms of attachment, invasion, and replication. Parasites were routinely passaged in human foreskin fibroblast (HFF) cells.

## Drug preparations

All drugs were purchased from Sigma (St Louis, MO). Both alendronate and clodronate were dissolved in RPMI media free of phenol red to form 5.4 mM stocks, and 8-quinoline and 5-chloro-8-quinolinol were dissolved in dimethyl sulfoxide (DMSO) to form 0.54 M stocks. Media prepared with mycophenolic acid (50 mg/ml) and xanthine (50 mg/ml) that inhibits the growth of Δ*hxgprt Toxoplasma* strains was used as a no growth control for assays performed using the RHΔ*hxgprt* GFP strain. All drug stocks were stored at 4°C.

## Fluorescence assays

Black 96-well tissue culture-treated plates with opaque bottoms were purchased from BD Falcon (Bridgeport, NJ). Each well was seeded with cells in a volume of 200 μl. Before infection, plates were washed with RPMI to remove all traces of phenol red, to reduce interference and to increase the sensitivity of fluorescence detection. Freshly lysed parasites were plated in parasite culture medium without phenol red at a concentration of $10^4$ parasites per well. Parasites were allowed to invade for 4 h before drug treatment was applied. Drug stocks were diluted three-fold through six wells, from 2.7 to 11 μM for alendronate, clodronate, and 5,5-dithiobis, and from 100 to 3.125 μM for 8-hydroxyquinoline and 5-chloro-8-quinolinol. Plates were kept in a humidified incubator at 37°C with 5% $CO_2$ and read in a Perkin-Elmer Wallac fluorescent plate reader. The following excitation (485 nm for both GFP and RFP) and emission (535 nm for GFP; 595 nm for RFP) values were used. Fluorescence readings were measured daily, the assay proceeded until day 7 post infection.

## Microscopy assays

Clear 96-well, Costar tissue culture-treated plates were purchased from Corning (Sigma-Aldrich, St Louis, MO). Cells were grown and infected as described above for the fluorescence assays. After 24 h of drug treatment, the average number of parasites present in ~100 vacuoles across 5–7 fields of view (1, 2, 4, 8, 16, and, in rare cases 32 parasites per vacuole) was calculated using a fluorescence microscope at ×32 magnification. Photos were also taken to show growth or inhibition visually.

## Quantitative PCR

Total RNA (2 μg) isolated from tachyzoites by the RNeasy mini kit (Qiagen) was reverse transcribed using random primers and Super-Script II (Invitrogen). Gene expression was measured by Taqman qPCR using an Applied Biosystems 7900HT Real-Time PCR System. The cycling program included 2 min at 50°C, 10 min incubation at 95°C followed by 40 cycles of 95°C for 15 s and 60°C for 1 min. Toxoplasma 18S rRNA and LDH1 were used as reference genes to normalize the quantity of transcripts (Livak and Schmittgen, 2001). Transcript levels were represented as $2^{-\Delta CT}$ to show absolute levels of transcript relative to every gene examined.

## Host cell viability assays

Host cell viability assays under drug treatment were performed using the CellTiter-Glo Luminescent Cell Viability Assay (Promega). The kit's reagent was prepared according to the protocol, then added to 96-well plates in which HFF cells had been left under drug treatment for either 24 h or 7 days. Plates were read using a Perkin-Elmer Wallac fluorescent plate reader. Data analysis was performed using Prism.

## Supplementary information

## Acknowledgements

This study was funded by the Canadian Institutes for Health Research (CIHR—MOP #84556 to JP and MEG) and the Natural Sciences and Engineering Research Council (NSERC #188266-04 to JP). JP also acknowledges support from the Ontario Ministry of Research and Innovation. MEG and MAC also acknowledge support from the Intramural Research Program of the National Institutes of Health (NIH) and National Institute of Allergy and Infectious Diseases (NIAID). JP is a member of the Center for the analysis of genome evolution and function (CAGEF). MEG is a Scholar of the Canadian Institute for Advanced Research (CIFAR) Program for Integrated Microbial Biodiversity. Computing resources were provided by the SciNet HPC Consortium.

*Author contributions:* JP and MEG conceived and designed the study. CS, JW, NN, and SSH performed reconstruction. CS and NN performed simulations. MAC performed drug assays. CS, JP, MAC, MEG, NN, and SSH analysed results. CS, JP, JW, MEG, MAC, NN, and SSH drafted manuscript.

## Conflict of interest

The authors declare that they have no conflict of interest.

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
