## [Review Process File · Molecular Systems Biology]

Metabolic Reconstruction Identifies Strain-Specific Regulation of Virulence in *Toxoplasma gondii*

Carl Song, Melissa A. Chiasson, Nirvana Nursimulu, Stacy S. Hung, James Wasmuth, Michael E. Grigg and John Parkinson

Corresponding author: John Parkinson, The Hospital for Sick Children

Review timeline:

Submission date:	10 May 2013
Editorial Decision:	21 June 2013
Revision received:	20 September 2013
Accepted:	10 October 2013

Editor: Maria Polychronidou

Transaction Report:

1st Editorial Decision

21 June 2013

Thank you again for submitting your work to Molecular Systems Biology. We have now heard back from the three referees who agreed to evaluate your manuscript. As you will see from the reports below, the reviewers acknowledged that your work addresses a potentially interesting topic. However, they raise a series of concerns, which should be carefully addressed in a revision of the manuscript.

Several of the reviewers' comments refer to the need to provide explanations and/or clarifications for several points throughout the manuscript. One of the more fundamental issues raised by reviewer #3, is that a sensitivity test regarding the approach to constrain fluxes, should be performed.

On a more editorial note, as reviewers #1 and #2 have pointed out, the manuscript has to be carefully re-written in order to become more concise and avoid repetitions.

If you feel you can satisfactorily deal with these points and those listed by the referees, you may wish to submit a revised version of your manuscript. Please attach a covering letter giving details of the way in which you have handled each of the points raised by the referees. A revised manuscript will be once again subject to review and you probably understand that we can give you no guarantee at this stage that the eventual outcome will be favorable.

Referee reports

Reviewer #1:

The reviewed paper reports the first large-scale metabolic network reconstruction for *Toxoplasma*

gondii, a pathogen of medium importance and moreover an important model organism for the research on other protozoans. The network is adequately verified by simulating the growth rates of different Toxo strains. As a possible application, single and double knockouts have been simulated aimed at the drug target prediction. Furthermore, some of the predictions have been experimentally tested, providing a degree of validation which is above the standard of other organism's first reconstructions.

Apparently, the presented network does not cover the full genomic information on metabolic reactions for Toxoplasma, and is considerably smaller than other comparable reconstructions. However, this is only a minor disadvantage as the model is capable to represent the most important cellular functions as it is. The only obstacle to the ready use of the network by other researchers is the somewhat insufficient annotation.

The paper is well written, the few exceptions are detailed below. The paper is relatively long but I consider it appropriate for the work covered. I recommend it for publication in MSB.

Major issues:

1. I highly recommend distributing the network also in SBML (current COBRA should be able to write the network in SBML). The Cytoscape project file holding the graphical information would also be helpful to the community.
2. Standard of documentation. The metabolites should be annotated with a standardized nomenclature such as CheBi (recommended) or KEGG (at least). I recommend an additional sheet in supplementary table 1 for metabolites (holding at least the information from the Metabolites table available at the project's web site). The reactions should be annotated with a nomenclature such as KEGG, reactome (as these resources have been used in the reconstruction anyway). References used (the Reactions file in the project's online version contains some) should be listed by reaction in the supplementary table. All this information should also be integrated in the SBML with an appropriate reporting standard such as MIRIAM.

Minor issues:

1. There is a contradiction between a sentence in the Results section where all organellar transport reactions are attributed to "passive diffusion" while in Materials I read "transporter proteins catalyzing intracellular transport reactions" which I interpret as active, furthermore "translocating ... accompanied by hydrolysis of ATP" which suggests an ATP-driven transport. For example, reaction "O61" is indeed an ATP-driven transport.
2. Results, first section, third paragraph. I did not understand the workflow represented in the sentence "Referencing the Kyoto ... metabolites (Figure 1B)." In which way KEGG was used to add what?
3. Discussion. third paragraph. It should be mentioned that it is also possible that reactions are missing from the network which would represent alternative routes.
4. Discussion. I would suggest to elaborate on what is still missing in the network (relevant pathways, compartments, level of detail), and an outlook in which ways it could be extended or refined.
5. Methods/Microscopy assays. Please indicate which cellular item is tagged by fluorescence.
6. The strategy to compile the initial reaction set was not clear from reading the manuscript. Was it either a complete set based on ToxoDB reactions, or was it only the necessary reactions to support growth and maintenance extracted from ToxoDB.
7. Legend to figure 5: Please explain unit "RFU". "(D)" unit " $2^{-\Delta Ct}$ ". Why not just $2^{-\Delta Ct}$, as I assume "Ct" is the number of amplification cycles. Delta refers to a difference, but a difference to what?

8. I recommend to change the order in the x axis of Figure 5(D), because the EC 2.7.1.90 is dealt with first (in subfigure A), then 4.1.2.13 (subfigure B).

9. Figure 1A. I understand that the numbers presented show the number of reactions drawn from the particular source. I would suggest to also show the total number of the reactions of the network contained in the particular resource. I wonder why BRENDA is mentioned here at all as it is not really used.

10. I would recommend to split Supplementary figure 3 in two as (B) and (C) only represent host cells.

11. Reannotation of genes TGME49_088450/Supplementary figure 1. It is not so straightforward to the reader why EC 1.5.1.12 is so preferable. Is it the position of the large peak at low values or the position of the small peak? The large yellow peak left means, for lot of genomes there is basically no sequence similarity - the quantity of nil does not matter. The small peak is very small and its position is not so different. Reannotation of genes TGME49_109730/Supplementary figure 1. This is much clearer. But the problem is that the similarity to EC 1.8.1.7 is also relatively good that it would be accepted as sufficient if the larger similarity to EC 1.8.1.9 did not exist. Please discuss this!

12. Especially in the final paragraph of the discussion, I found some statements and phrases duplicated. The general description of the network appears in the Abstract, Introduction, Results, and Discussion, and I suggest to shorten the respective paragraphs to reduce the overlap.

Language issues:

1. Abstract. "are driven altered capacities": "by" missing.
2. Introduction "parasite strain `Type` ...": quote sign type.
3. Introduction "predicator" -> "predictor"
4. Introduction "offering a putative role": putative does not fit well in the context, maybe potential?
5. Introduction "reveal identical enzyme complements" Although I can guess what is meant, I do not think the phrase "identical enzyme complements" (used several times throughout the manuscript) covers that appropriately. I suggest "enzymes with the same catalytic activity" instead.
6. Introduction. "the impact of differential expression on ... " There is some confusion in the semantic layers here. The differential expression is not the acting entity. The cells express their genes, and feature a certain growth rate, and that differs between the strains.
7. Results "associated with other gene annotations" doubly, "with other genes" is enough
8. Results. "different cell locations as previous" -> "previously"
9. Results. "glucose-6 phosphate isomerise" -> "glucose-6-phosphate isomerase"

Issues in the files downloadable at the project's website:

1. The references in the Reactions table should be given with full citation, not only Author name and year (a separate document/sheet listing the full citations is recommended).

Technical issues:

It is difficult to place comments without page numbering. The page breaks in the pdf and the docx are not identical. Therefore, I refrained from giving more detailed text positions for the comments. Hopefully, text search will disclose which text fragments I'm referring to.

Reviewer #2:

Song et al describe in their manuscript a metabolic reconstruction of *Toxoplasma gondii* and the analysis of specific metabolic properties of different strains, which they then exploit to identify potential drug targets. The authors assemble the manually curated metabolic reconstruction using current reconstructions techniques and standards. To model strain-specific differences, the authors then integrated mRNA expression data from four strains as flux constraints for the model and

performed flux balance analysis. Single and double gene knockouts were then simulated and the results were compared with published information. The authors identified 38 potential drug targets that lack homologs in human. By comparing predictions for the four strains, as well as model predictions with a published reconstruction of *Plasmodium falciparum*, species- and strain-specific differences were identified. Finally, the authors validate three of the potential drug targets in vitro thus demonstrating the correctness of their predictions and the model's predictive potential. While this study is interesting and insightful, it uses established methods in the field. The reviewer feels that, albeit important, this work may be more suitable for a more specialized journal.

- While the manuscript is well written it is repetitive in parts and should be streamlined.
- The reconstruction seems comparatively small with only 380 gene annotations, compared with, for example, the *P. falciparum* reconstruction with 579 gene annotations. The authors may need to discuss this point more (e.g. pathways not captured by the reconstruction).
- The definition of 'dead end reactions' is incorrect.
- It is not clear to the reviewer how the mRNA expression data were applied as constraints. For instance, have the constraints been applied to lower and upper bounds on each reaction? If so how did you treat reversible reactions? The authors should add a more detailed description to the method section.
- Moreover, the authors state that the flux constraints were scaled such that the model would predict the expected doubling time. Consequently, the authors state that they predict the doubling time correctly, which is expected if the models' constraints were scaled in such manner. Please clarify which models were scaled and which ones.
- It is not clear whether the analysis of active pathways refers to the mapped data or to simulation results. Please clarify. If it refers to mapped data please also state what one could conclude by using the model rather than the gene expression data alone. If it refers to simulations please clarify if alternate optimal solutions have been considered.
- On page 38, Figure 5 is labeled as Figure 4, while Figure 4 on page 37 is unlabeled.

Reviewer #3:

This paper describes the metabolic reconstruction of *T. gondii* and use of the reconstructed metabolic network for identification of strain specific variances in terms of impact on host cells. Furthermore, genetic variations are used together with the metabolic reconstruction to evaluate different drug targets. Even though this paper does not provide new conceptual approaches it is an impressive illustration of how genome-scale metabolic modeling can be used to evaluate host-parasite interaction, and further to evaluate different drug targets. In particular of interest is the coupling of the modeling with different genetic variants and this represent one of the first illustrations of individualized models for drug discovery.

The paper is very well written and I have no comments to the method, concept or results., except for the approach to constrain fluxes. The authors uses transcription data to constrain fluxes. This has been tried before and they chose a relatively conservative approach that will probably give reasonable results. However, I strongly suggest that the authors perform a sensitivity test, in particular to their assumption of scaling maximum fluxes to the the percentage of gene expression to maximum (an example of 20% is given in the M&M). It would be relatively easy to evaluate if the overall conclusions were affected by performing variations in this upper bound for key fluxes (particularly those that were constrained to very low values - I guess that many fluxes were not de-facto constrained).

Response to Reviewers

Reviewer #1:

*The reviewed paper reports the first large-scale metabolic network reconstruction for *Toxoplasma gondii*, a pathogen of medium importance and moreover an important model organism for the research on other protozoans. The network is adequately verified by simulating the growth rates of different *Toxo* strains. As a possible application, single and double knockouts have been simulated aimed at the drug target prediction. Furthermore, some of the predictions have been experimentally tested, providing a degree of validation which is above the standard of other organism's first reconstructions.*

*Apparently, the presented network does not cover the full genomic information on metabolic reactions for *Toxoplasma*, and is considerably smaller than other comparable reconstructions. However, this is only a minor disadvantage as the model is capable to represent the most important cellular functions as it is. The only obstacle to the ready use of the network by other researchers is the somewhat insufficient annotation.*

The paper is well written, the few exceptions are detailed below. The paper is relatively long but I consider it appropriate for the work covered. I recommend it for publication in MSB.

We would like to thank the reviewer for their highly positive and encouraging comments. As the reviewer appreciates, in constructing the model we choose to be relatively conservative concerning the choice of reactions to include. At the same time, we believe the model will form the basis for subsequent refinements and as such we have improved the annotation of the network as suggested below.

Major issues:

1. I highly recommend distributing the network also in SBML (current COBRA should be able to write the network in SBML). The Cytoscape project file holding the graphical information would also be helpful to the community.

The SBML file has now been generated and is included as a downloadable resource, along with the Cytoscape file, both of which will be made available through ToxoDB as well as our own website. Text has been added to indicate the availability of the SBML file (Page 17 lines 23-26).

2. Standard of documentation. The metabolites should be annotated with a standardized nomenclature such as CheBi (recommended) or KEGG (at least). I recommend an additional sheet in supplementary table 1 for metabolites (holding at least the information from the Metabolites table available at the project's web site). The reactions should be annotated with a nomenclature such as KEGG, reactome (as these resources have been used in the reconstruction anyway). References used (the Reactions file in the project's online version contains some) should be listed by reaction in the supplementary table. All this information should also be integrated in the SBML with an appropriate reporting standard such as MIRIAM.

In line with these standards we have chosen to use the KEGG nomenclature as this provides both standardized reaction ID's as well as metabolite ID's. Supplemental Table 1 now contains references to KEGG defined reaction ID's, and the reactions themselves are now defined with KEGG defined compound ID's for the metabolites (this also includes Table 3). The references that were used in constructing the model are now included as PubMed ID's next to the reaction. Note, not all reactions have PubMed ID's as they may be included either as gene predictions or as gap filling reactions. Associated with these changes, we also provide two additional supplemental files: Supplemental Table 5 is a list of mappings of KEGG compound ID's to metabolites; and Supplemental Table 6 provides details of the citations used to assign reactions (these were previously provided as Supplemental Material). This information is now captured in the SBML file as suggested using the MIRIAM reporting standard as suggested. This includes the inclusion of consistent meta ID's in the form of uniform resource name to define compounds, reactions, pubmed ID's and gene annotations.

Minor issues:

1. *There is a contradiction between a sentence in the Results section where all organellar transport reactions are attributed to "passive diffusion" while in Materials I read "transporter proteins catalyzing intracellular transport reactions" which I interpret as active, furthermore "translocating ... accompanied by hydrolysis of ATP" which suggests an ATP-driven transport. For example, reaction "O61" is indeed an ATP-driven transport.*

We thank the reviewer for noting this contradiction; organellar transport reactions include both active transporters as well as those involving passive diffusion. We have modified the text both in Results (Page 5 lines 37-40) and Methods (Page 18 lines 19-21) to clarify this issue.

2. *Results, first section, third paragraph. I did not understand the workflow represented in the sentence "Referencing the Kyoto ... metabolites (Figure 1B)." In which way KEGG was used to add what?*

Enzymatic reactions, as defined by EC identifiers, were cross-referenced with the Kyoto Encyclopedia of Genes and Genomes (KEGG) database (Kanehisa et al, 2006), to obtain reaction details including substrates and product metabolites, as well as reaction stoichiometry and direction. Text providing this clarification is now provided in Results (Page 6 lines 6-11).

3. *Discussion. third paragraph. It should be mentioned that it is also possible that reactions are missing from the network which would represent alternative routes.*

We agree with the reviewer and have included this possibility in the Discussion (Page 14 lines 38-41)

4. *Discussion. I would suggest to elaborate on what is still missing in the network (relevant pathways, compartments, level of detail), and an outlook in which ways it could be extended or refined.*

As suggested, we have significantly extended this section in the discussion and highlight additional future improvements including: additional compartments, pathways (glycan metabolism and arachidonic acid) as well as capturing kinetic data to aid in the assignment of reaction constraints. (Page 17 lines 1-20).

5. *Methods/Microscopy assays. Please indicate which cellular item is tagged by fluorescence.*

Parasites with stable integration of the GFP or RFP cassette were selected by fluorescence. The point of insertion is unknown, but the parasites are identical in phenotype to wild-type in terms of attachment, invasion, and replication. This text has now been added to methods (Page 20 lines 5-8)

6. *The strategy to compile the initial reaction set was not clear from reading the manuscript. Was it either a complete set based on ToxoDB reactions, or was it only the necessary reactions to support growth and maintenance extracted from ToxoDB.*

The initial reaction set was compiled from the complete set of EC annotations provided by ToxoDB and BRENDA, supplemented by DETECT predictions generated in house. Segregated reactions that do not link to other parts of the network and are responsible for the production of non-essential metabolites or whose substrates are not considered metabolites (e.g. polypeptides or other polymers) were removed. Text to this effect has been added to Methods to clarify this (Page 18 lines 6-8).

7. *Legend to figure 5: Please explain unit "RFU". "(D)" unit "2^{-delta} Ct". Why not just 2^{-Ct}, as I assume "Ct" is the number of amplification cycles. Delta refers to a difference, but a difference to what?*

RFU stands for relative fluorescent units, a standard unit of measurement for fluorescence. Delta Ct refers to the fact that we are normalizing transcription levels for genes of interest to a reference gene that is highly and uniformly expressed (a 'housekeeping gene'); in our study we

used the Toxoplasma 18S rDNA gene array as this reference. For all genes of interest, we subtracted the 18S Ct value from the gene of interest's Ct value to correct for any variation in initial cDNA concentrations. The figure legend has been updated as suggested (Page 31 lines 3-5) and we now include a reference for this in Methods (Page 21 lines 1-4).

8. *I recommend to change the order in the x axis of Figure 5(D), because the EC 2.7.1.90 is dealt with first (in subfigure A), then 4.1.2.13 (subfigure B).*

The order in the x axis has now been changed as requested.

9. *Figure 1A. I understand that the numbers presented show the number of reactions drawn from the particular source. I would suggest to also show the total number of the reactions of the network contained in the particular resource. I wonder why BRENDA is mentioned here at all as it is not really used.*

While the BRENDA database does not provide data beyond our own literature curation (other literature evidence), it is nonetheless a well respected resource that we have chosen to include, to demonstrate both support for the 35 enzymes that it captures, as well as to highlight its current limitations in terms of coverage. We now include the total number of reactions provided by each resource as suggested.

10. *I would recommend to split Supplementary figure 3 in two as (B) and (C) only represent host cells.*

Supplemental Figure 3 has now been split into Supplemental figures 3 and 4 as requested.

11. *Reannotation of genes TGME49_088450/Supplementary figure 1. It is not so straightforward to the reader why EC 1.5.1.12 is so preferable. Is it the position of the large peak at low values or the position of the small peak? The large yellow peak left means, for lot of genomes there is basically no sequence similarity - the quantity of nil does not matter. The small peak is very small and its position is not so different.*

We appreciate this concern, and agree that the reason for the reannotation of TGME49_088450 would benefit from additional explanation. In the plot for EC 1.2.1.3, most hits (represented by the tall peak) are very low-scoring (close to having an almost non-significant bit alignment score) and are slightly biased to the negative hits. A minority of hits (small peak on the far right) overlap with a very small portion of 1.2.1.3 positive hits. However, given the majority of the positive hit distribution occurs around a bit score of ~400 and ~800, it is not surprising that EC 1.2.1.3 obtains a comparably lower probability score (implying that TGME49_088450 is less likely to belong to EC 1.2.1.3 enzymes). In the plot for EC 1.5.1.12, again, most hits are very low-scoring and biased towards negative hits; however, a minority (small peak on far right) overlap with a much larger section of the positive hit distribution (other large section of the positive hit distribution occurs around a bit score of ~600). This contributes to a higher probability score and suggests that TGME49_088450 is more likely to belong to the family of enzymes annotated as EC 1.5.1.12. We have added further explanation to the legend of Supplemental Figure 1 (Page 36 lines 4-32).

Reannotation of genes TGME49_109730/Supplementary figure 1. This is much clearer. But the problem is that the similarity to EC 1.8.1.7 is also relatively good that it would be accepted as sufficient if the larger similarity to EC 1.8.1.9 did not exist. Please discuss this!

Again we agree that this figure benefits from additional explanation.

In the plot for EC 1.8.1.9, the majority of the hits to EC 1.8.1.9 enzymes (i.e. tall peak) are very high scoring (> 1000 bit score) and overlap with a region of the positive hit distribution for EC 1.8.1.9. The small peak in this plot is essentially uninformative since it corresponds to very low scoring hits and is ambiguous in its contribution to the positive or negative hit distribution. These aspects of the plot help to explain the higher probability score, and therefore greater likelihood that TGME49_109730 belongs to the family of enzymes annotated as EC 1.8.1.9.

In the plot for EC 1.8.1.7, the majority of hits for TGME49_109730 (tall peak) are of relatively moderate/low score (~650 bit score) with some overlap to both positive and negative hit distributions, but with a slight bias towards the positive hit distribution. Together, the lack of very high-scoring hits (as is the case with EC 1.8.1.9) and overlap with both positive and negative hit distributions results in a lower probability score (less likely to belong to enzymes annotated as EC 1.8.1.7). We have added further explanation to the legend of Supplemental Figure 1 (Page 36 lines 4-32).

12. Especially in the final paragraph of the discussion, I found some statements and phrases duplicated. The general description of the network appears in the Abstract, Introduction, Results, and Discussion, and I suggest to shorten the respective paragraphs to reduce the overlap.

We have endeavored to reduce the amount of duplication and overlap in the various sections and have considerably shortened the respective paragraphs as suggested – these paragraphs include: Page 4, lines 16-18; Page 1, lines 3-10; Page 14 lines 2-7, 13-14, 20-22; Page 16 lines 9-12; and Page 17 lines 24-29.

Language issues:

1. Abstract. "are driven altered capacities": "by" missing.

Corrected as indicated (Page 2 line 8).

2. Introduction "parasite strain `Type` ...": quote sign type.

Corrected as indicated (Page 3 line 12).

3. Introduction "predicator" -> "predictor"

Corrected as indicated (Page 3 line 12).

4. Introduction "offering a putative role": putative does not fit well in the context, maybe potential?

Corrected as indicated (Page 3 line 24).

5. Introduction "reveal identical enzyme complements" Although I can guess what is meant, I do not think the phrase "identical enzyme complements" (used several times throughout the manuscript) covers that appropriately. I suggest "enzymes with the same catalytic activity" instead.

In the absence of detailed kinetic data, we think it is not possible to state that the equivalent enzymes have the same catalytic activities. However we do note the prior ambiguity and have changed the sentence to “While genome comparisons reveal identical sets of genes encoding enzymes with the same predicted functional roles across the three strains...” (Page 3 lines 24-25).

6. Introduction. "the impact of differential expression on ... " There is some confusion in the semantic layers here. The differential expression is not the acting entity. The cells express their genes, and feature a certain growth rate, and that differs between the strains.

We have now changed this sentence to “However, what is not known is how the differential expression of these genes across different *Toxoplasma* strains may influence their growth potential and hence virulence.” (Page 3 lines 26-28).

7. Results "associated with other gene annotations" doubly, "with other genes" is enough

Corrected as indicated (Page 5 line 17).

8. Results. "different cell locations as previous" -> "previously"

This sentence has been slightly revised to address minor point 1 above (Page 5 lines 37-40).

9. Results. "glucose-6 phosphate isomerise" -> "glucose-6-phosphate isomerase"

Corrected as indicated (Page 10 line 36).

Issues in the files downloadable at the project's website:

1. The references in the Reactions table should be given with full citation, not only Author name and year (a separate document/sheet listing the full citations is recommended).

As suggested, the references in the reactions table are now given PubMed ID's with the full citation cross-referenced in Supplemental Table 6.

Reviewer #2:

Song et al describe in their manuscript a metabolic reconstruction of Toxoplasma gondii and the analysis of specific metabolic properties of different strains, which they then exploit to identify potential drug targets. The authors assemble the manually curated metabolic reconstruction using current reconstructions techniques and standards. To model strain-specific differences, the authors then integrated mRNA expression data from four strains as flux constraints for the model and performed flux balance analysis. Single and double gene knockouts were then simulated and the results were compared with published information. The authors identified 38 potential drug targets that lack homologs in human. By comparing predictions for the four strains, as well as model predictions with a published reconstruction of Plasmodium falciparum, species-and strain-specific differences were identified. Finally, the authors validate three of the potential drug targets in vitro thus demonstrating the correctness of their predictions and the model's predictive potential. While this study is interesting and insightful, it uses established methods in the field. The reviewer feels that, albeit important, this work may be more suitable for a more specialized journal.

We appreciate the reviewer's candid opinion, however we feel that the findings would be of interest to a widespread audience. This is one of the first studies to use genome-scale metabolic modeling to investigate strain-specific differences in growth rates and susceptibilities to targeted therapeutic intervention. In addition to demonstrating the need to consider strain variation during the development of novel anti-parasitic therapeutics, this work introduces a new evolutionary model that proposes the regulation of growth potential may help drive the broadening of a pathogen's host range.

- While the manuscript is well written it is repetitive in parts and should be streamlined.

As indicated above, we have considerably shortened the respective paragraphs as suggested – these paragraphs include:

Page 4, lines 16-18; Page 1, lines 3-10; Page 14 lines 2-7, 13-14, 20-22; Page 16 lines 9-12; and Page 17 lines 24-29.

- The reconstruction seems comparatively small with only 380 gene annotations, compared with, for example, the P. falciparum reconstruction with 579 gene annotations. The authors may need to discuss this point more (e.g. pathways not captured by the reconstruction).

We acknowledge the conservative nature of the iCS382 reconstruction (note during the revision process we have undergone some further refinement of the model which, although did not significantly alter the results, did lead to the inclusion of two further enzyme encoding genes – hence the change to iCS382 see further comments in response to Reviewers 3 comments). Our strategy for this reconstruction was to include only pathways for which there was significant supporting evidence (e.g. presence of several reactions as determined through the four resources used for the construction). As also recommended by Reviewer 1 (comment 4) we have added additional discussion regarding additional reactions and pathways that future iterations of the model may wish to address (Page 17 lines 1-20).

- The definition of 'dead end reactions' is incorrect.

'Dead end reactions' involve metabolites that are neither produced nor consumed by other metabolic reactions in the network and additionally lack evidence supporting the import or export of the metabolites implying that the flux for the reaction must be zero. Text to this effect has now been included in the text (Page 6 lines 3-6).

- It is not clear to the reviewer how the mRNA expression data were applied as constraints. For instance, have the constraints been applied to lower and upper bounds on each reaction? If so how did you treat reversible reactions? The authors should add a more detailed description to the method section.

For reversible reactions, the reviewer is correct that constraints were applied equally to upper and lower bounds. For example, a reversible reaction encoded by a gene expressed at 20% of the level of the most highly expressed gene received constraints of -200 to 200 mmol/g_{DW}·h. For irreversible reactions, the lower bound was set to 0. On the other hand, an irreversible reaction encoded by a gene expressed at 20% of the level of the most highly expressed gene received constraints of -0 to 200 mmol/g_{DW}·h. Text clarifying the implementation of these constraints is now provided in Methods (Page 19 lines 9-15).

- Moreover, the authors state that the flux constraints were scaled such that the model would predict the expected doubling time. Consequently, the authors state that they predict the doubling time correctly, which is expected if the models' constraints were scaled in such manner. Please clarify which models were scaled and which ones.

We agree that this was not clear in the original text. The calculation of scaling factors was performed only once, for the Me49 expression data to derive a doubling time of 11.8 hours. The same scaling that was derived from the Me49 model was then applied to the other three strains expression data agnostic of their doubling time. Hence the predicted doubling time for these other strains is an emergent rather than expected finding. We have added text to the Methods section to help clarify (Page 19 lines 18-20).

- It is not clear whether the analysis of active pathways refers to the mapped data or to simulation results. Please clarify. If it refers to mapped data please also state what one could conclude by using the model rather than the gene expression data alone. If it refers to simulations please clarify if alternate optimal solutions have been considered.

We believe this comment concerns the explanation regarding the observed strain differences in growth rate (Page 7 lines 40-43). We agree that there is some ambiguity in this section. The predicted growth rates for the three strains (RH, GT and Prugniaud) were obtained by applying the same scaling to their respective expression datasets as was used to obtain a doubling time of 11.8 hours for strain Me49. Given that this was the only difference in the strain models, we explored the pathways that appeared most differentially expressed between Me49 and RH as a potential explanation regarding the observed changes in growth rate. In subsequent sections of the manuscript we describe how we use the model to investigate reaction knockouts which corroborate the suggestion that the differences in growth rate is largely due to changes in expression of enzymes involved in energy production. We have now modified this section to be more explicit concerning the mapped data (Page 7 lines 40-43).

- On page 38, Figure 5 is labeled as Figure 4, while Figure 4 on page 37 is unlabeled.

This may have been an error that crept in during the upload of the figures, we will carefully work with the production team to ensure that the labeling of the figures is correct.

Reviewer #3:

*This paper describes the metabolic reconstruction of *T. gondii* and use of the reconstructed metabolic network for identification of strain specific variances in terms of impact on host cells. Furthermore, genetic variations are used together with the metabolic reconstruction to evaluate different drug targets. Even though this paper does not provide new conceptual approaches it is an impressive illustration of how genome-scale metabolic modeling can be used to evaluate host-*

parasite interaction, and further to evaluate different drug targets. In particular of interest is the coupling of the modeling with different genetic variants and this represent one of the first illustrations of individualized models for drug discovery.

The paper is very well written and I have no comments to the method, concept or results, except for the approach to constrain fluxes. The authors uses transcription data to constrain fluxes. This has been tried before and they chose a relatively conservative approach that will probably give reasonable results. However, I strongly suggest that the authors perform a sensitivity test, in particular to their assumption of scaling maximum fluxes to the percentage of gene expression to maximum (an example of 20% is given in the M&M). It would be relatively easy to evaluate if the overall conclusions were affected by performing variations in this upper bound for key fluxes (particularly those that were constrained to very low values - I guess that many fluxes were not de-facto constrained).

This is an excellent comment and we have performed sensitivity analyses as recommended to investigate the robustness of the results. As the reviewer notes, there were in fact 324 of 571 which were not constrained (or rather were set to the maximum range allowed in the model – a conservative approach which provides the greatest flexibility in flux distributions). To examine the impact of changing the upper bounds of individual reactions as suggested we performed a systematic set of simulations. Working with the constraints that we defined for the model based on strain Me49 (baseline model), we altered the upper bound of the constraint to the maximum allowed in the model for each of the 400 metabolic reactions. We then examined the impact of each of these changes on the single reaction knockout predictions (i.e. 400 sets of single knockout predictions) and compared them against the baseline model. As previous we defined reaction knockouts as having one of four effects (Figure 3B): Essential, Major (growth rate <80% of baseline), Minor (growth rate 80-99% of baseline) or no effect. For 396 of the 400 reactions, raising the upper bound had no impact on the classification of single knockouts. Only for four reactions did altering the upper bound to maximum have an impact: ubiquinol-cytochrome-c reductase (EC:1.10.2.2), phosphoglycerate kinase in the cytosol (EC:2.7.2.3a), phosphoglycerate kinase in the apicoplast (EC:2.7.2.3b) and 6-phosphogluconolactonase (EC:3.1.1.31) which resulted in 38, 24, 27 and 25 category changes respectively (Supplemental Figure 3A). Most of these were subtle i.e. from Essential to Major (10 changes), Major to Minor (7 changes), Minor to No effect (75 changes). All four were identified as bottleneck reactions (Supplemental Figure 3B) and are involved in energy production, again highlighting the overall impact of flux changes in energy production on growth rate. Nevertheless, given the relatively modest changes, we conclude that the model is, for the most part, robust to the maximum constraint assignments.

Next we were interested in examining if the scaling applied to individual reactions based on the expression data had a significant impact on the respective growth rates of strains RH and Me49. Given the importance of energy metabolism, we therefore selected five reactions involved in energy production pathways as well as the three reactions targeted in our drug inhibition assays (Triose-phosphate isomerase - EC: 5.3.1.1a; Glucose-6-phosphate isomerase – EC:5.3.1.9; Glyceraldehyde-3-phosphate dehydrogenase – EC1.2.1.12a; Pyruvate kinase – EC:2.7.1.40a; Ribulose-phosphate 3 epimerase – EC: 5.1.3.1; coproporphyrinogen oxidase – EC: 1.3.3.3; fructose-bisphosphate aldolase – EC: 4.1.2.13; and diphosphate-fructose-6-phosphate 1-phosphotransferase – EC: 2.7.1.90) and examined the impact of decreasing their maximum flux constraint on growth rate (Supplemental Figure 3B). Note if a reaction was reversible, the constraint was similarly decreased for both the upper and lower bounds. For the three reactions, targeted for drug assay validation, we note that only by decreasing the maximum constraint to 10% of that assigned in the model, do we begin to see an impact on growth rate. On the other hand, reducing Glucose-6-phosphate isomerase (EC:5.3.1.9) and Pyruvate kinase (EC:2.7.1.40a) to ~80% of the constraint assigned in the model does impact predicted growth rate. This suggests that the model may be sensitive to the accuracy of the constraints applied to these reactions, and again emphasizes the importance of regulation of energy production pathways. At the same time, we note that for all reactions, the decrease in growth rate with respect to decreasing constraint results in a shallower gradient for strain RH compared to strain Me49, consistent with one of the main findings of our study that strain Me49 is predicted to be more sensitive to inhibition of enzymes involved in energy production pathways. It should also be noted that the constraints we have used in these analyses are not limiting growth rate.

Given these analyses, we conclude that our overall predictions remain robust to changes in the constraints assigned to individual enzymes; observed strain differences appear to be driven by relative differences in expression in global enzyme expression across the entire set of energy production pathways. We have included a section on these sensitivity analyses in the Results (Page 11 lines 14-26 and Supplemental Figure 3).

Finally it is worth reporting that during revision of this manuscript, we noted some areas of the model that would benefit from further refinement. This resulted in following changes: (1) Five new enzymatic reactions have been added (2.3.1.16b, 2.7.7.18, 3.1.3.5j, 3.1.3.5k, 3.5.4.12) – two of which had new gene associations and were thus responsible for the change in model name from iCS380 to iCS382; (2) The biomass equation was changed to include geranylgeranyldiphosphate; (3) Four organellar transport reactions were added for the porphyrin subsystem to allow correct transport between departments (O82, O83, O84, O85); (4) The organellar transport reaction shuttling Fe²⁺ between cytosol and apicoplast was changed to be between cytosol and mitochondrion; (5) A diffusion reaction for ethanol was added; (6) A sink reaction for acyl carrier protein was changed to a demand reaction; and (7) New sink reactions were added for glycolaldehyde and 3,4-dihydroxy-L-phenylalanine. The manuscript has been updated to note these changes. However despite these changes, we noted only subtle differences in our results. Growth rates were relatively unperturbed (iCS380 = 2.9 hours for RH; iCS382 = 3.0 hours for RH), the number of essential enzymes changed from 252/498 to 242/507, while strain differences were very similar to the previous model (e.g. 8 of the top 9 reaction knockouts that display the greatest difference on growth rates between strain RH and Me49 involve the same glycolytic enzymes). Hence our previously reported findings have remained robust to these model refinements.

Acceptance letter

10 October 2013

Thank you again for sending us your revised manuscript. We are now satisfied with the modifications made and I am pleased to inform you that your paper has been accepted for publication.

Thank you very much for submitting your work to Molecular Systems Biology.

Reviewer #1:

Recommended for publication as it is.